



# Effect of humidity on the first steps of atmospheric new particles formation: Computational study of hydrated molecular clusters

Ivo Neefjes[1], Yosef Knattrup[1], Haide Wu[1], Georg Baadsgaard Trolle[1], Jonas Elm[1], and Jakub Kubečka[1]

[1]Department of Chemistry, Aarhus University, Langelandsgade 140, 8000 Aarhus C, Denmark

**Correspondence:** Jakub Kubečka (ja-kub-ecka@chem.au.dk)

**Abstract.** To improve computational modeling of hydrated atmospheric molecular clusters, we systematically evaluated quantum-chemical methods for predicting accurate structural and energetic properties of clusters containing a variety of atmospherically relevant acids and bases, with up to five water molecules. We find that the commonly applied $\omega$B97X-D/6-31++G(d,p) method with DLPNO$^{NormalPNO}$–CCSD(T$_0$)/aug-cc-pVTZ electronic energy correction is suitable for hydrated clusters. Composite density functional methods such as B97-3c, r$^2$SCAN-3c and $\omega$B97X-3c are effective for pre-screening or modeling large clusters, while the local natural orbital approach LNO–CCSD(T)/aug′-cc-pVTZ is well-suited for accurate refinement due to its low memory requirements, high accuracy, and favorable computational scaling. Nevertheless, the $\omega$B97X-3c method has a reasonable accuracy even without the electronic energy correction.

We also assessed thermochemical corrections beyond the conventional harmonic oscillator approximation applied only to the lowest free-energy structure. For the limiting cases of no corrections and the ideal maximum corrections, we calculated hydration distributions and particle formation rates, with a specific emphasis on sulfuric acid–ammonia (SA–AM), sulfuric acid–dimethylamine (SA–DMA), and methanesulfonic acid–methylamine (MSA–MA) clusters. Hydration of small clusters is generally limited, with only selected SA- and MSA-containing clusters showing substantial hydration. Due to the high water concentration in the atmosphere, hydration equilibrates fast, increasing the number of accessible states, and thus stabilizing clusters. However, its effect on cluster formation and new particle formation is highly system dependent.

MSA–MA particle formation rates are more sensitive to hydration than those of SA–AM or SA–DMA, though the enhancement remains modest. Despite being more hydrated than SA–DMA clusters, MSA–MA clusters form new particles at relatively low rates, comparable to SA–AM. Under typical atmospheric conditions, SA–DMA is expected to dominate new particle formation, even at high humidity.

## 1 Introduction

Aerosol particles—solid and liquid particles suspended in the atmosphere—significantly influence both global climate (Li et al., 2022) and human health (Falcon-Rodriguez et al., 2016; Mei et al., 2018). While some aerosols are emitted directly from sources like sea spray, desert dust, volcanic eruptions, pollen, and fossil fuel combustion, most are formed in the atmosphere through a gas-to-particle conversion process known as new particle formation (NPF) (Kulmala et al., 2013). In NPF, low-volatility gas-phase molecules collide and stick together to form atmospheric molecular clusters. These clusters can continue to





grow into aerosol particles through condensation and coagulation. Aerosol particles impact the climate directly, by scattering incoming solar radiation, and indirectly, by providing a surface onto which water can condense to form clouds (Haywood and Boucher, 2020; Lohmann and Feichter, 2005). The latest IPCC assessment report indicates that aerosol particles are responsible for the largest uncertainty in current climate models (Intergovernmental Panel on Climate Change (IPCC), 2023).

This uncertainty is mainly due to limited knowledge about the early stages of NPF, where gas-phase molecules form clusters of $\sim$2 nm in diameter (Tröstl et al., 2016).

While the full range of atmospheric molecules contributing to NPF is still unknown, research has shown that clusters containing various acids and bases can rapidly form under atmospheric conditions. Acid–base clusters are stablized by proton transfer between the acid and base components, forming strongly bound salt. Sulfuric acid ($H_2SO_4$; SA; Sipilä et al. (2010))

plays a well-established role in NPF, while other acids, such as methanesulfonic acid ($CH_3SO_3H$; MSA; Dawson et al. (2012)) and nitric acid ($HNO_3$; NTA; Wang et al. (2020)) have been proposed as potential contributors. Formic acid (HCOOH; FA) and acetic acid ($CH_3COOH$; ACA), the most common organic acids in the atmosphere (Andreae et al., 1988; Keene et al., 1983; Keene and Galloway, 1984; Galloway et al., 1982; Millet et al., 2015), have been shown in computational studies to enhance NPF (Zhang et al., 2022). The most widely studied bases in atmospheric acid–base clusters are amines, with dimethylamine

(($CH_3)_2NH$; DMA) and trimethylamine (($CH_3)_3N$; TMA) playing a significant role, while ammonia ($NH_3$; AM), methylamine ($CH_3NH_2$; MA), and ethylenediamine ($C_2H_4(NH_2)_2$; EDA) have lower contributions (Almeida et al., 2013; Kurtén et al., 2008; Jen et al., 2016; Myllys et al., 2019; DePalma et al., 2012, 2014; Kirkby et al., 2011).

Water ($H_2O$; W) is ubiquitous in the atmosphere. At high relative humidities (RH), its concentration can reach $\sim 10^{17}$ cm$^{-3}$, about 10 orders of magnitude higher than that of particle-forming vapors such as sulfuric acid and bases. While water molecules

cannot form pure water clusters on their own under typical atmospheric conditions, they can participate in the formation of clusters with other atmospheric molecules (Carlsson et al., 2020). Several atmospheric measurement studies have investigated the effect of RH on NPF, generally finding an anticorrelation between NPF rates and RH (Birmili and Wiedensohler, 2000; Birmili et al., 2003; Boy and Kulmala, 2002; Laaksonen et al., 2008; Woo et al., 2001). Conversely, controlled laboratory studies indicate that increased RH can positively influence particle formation rates (Duplissy et al., 2016; Merikanto et al.,

2016). This discrepancy is believed to result from the fact that, although higher RH can directly boost NPF rates, it can also have indirect effects—like increasing cloud cover—that might reduce NPF in the atmosphere as lowered solar radiation leads to reduced gas-phase oxidation chemistry and the hygroscopic growth of preexisting particles increases the overall condensation sink (CS) factor (Hamed et al., 2011). However, it remains unclear whether water induces a consistent shift in NPF rates or affects them in more complex, condition-dependent ways.

State-of-the-art experimental techniques, such as condensation particle counters and particle size magnifiers, can detect aerosol particles down to sizes of $\sim$1.5–3 nm (McMurry, 2000; Vanhanen et al., 2011). However, these methods provide limited information on the chemical composition of the detected particles. While chemical ionization mass spectrometers (CIMS; Zapadinsky et al. (2019); Passananti et al. (2019); Jokinen et al. (2012)) offer molecular insights into clusters, fragmentation artifacts often distort cluster populations, complicating the characterization of sub-2–3 nm particles. In recent decades,

computational chemistry methods have been extensively used to address this challenge (Elm et al., 2020). Numerous studies





have focused on pure water clusters, exploring and characterizing their potential energy surface as well as examining their properties such as energetics (e.g., binding energies, HOMO-LUMO gap, and vibrational spectra), geometry of molecular interaction, charge distribution, and dipole moments (Gao et al., 2022; Andersson, 2023; Nguyen et al., 2008; Tribello et al., 2011; García-Argote et al., 2024; Nandi et al., 2021). Accurate modeling of sub-2–3 nm hydrated clusters remains computa-
tionally demanding, as they can consist of tens of molecules and include a variety of possible molecular species, necessitating the use of approximations. Hence, in computational studies of atmospheric molecular clusters, water is often excluded to reduce computational costs. This exclusion is based on the assumption that experimental studies are typically conducted under similar relative humidity (RH) conditions, minimizing systematic errors from neglecting water. Nevertheless, several computational studies have specifically investigated hydrated clusters, highlighting the potential role of water in aerosol formation. Ianni and
Bandy (2000) combined computational chemistry and classical thermodynamics to examine the hydration distributions of SA monomers and dimers. Kurtén et al. (2007) extended this work to SA–AM clusters, while Henschel et al. (2014, 2016) explored the role of humidity in SA–AM and SA–DMA nucleation. These studies demonstrated that water influences proton transfer in atmospheric acid–base clusters and can either promote or inhibit particle formation rates, depending on the cluster composition and environmental conditions. Similar findings were later reported by Ge et al. (2020), Myllys et al. (2021), and Myllys (2023).
With the growing number of potential NPF precursor candidates, multiple studies have investigated the hydration of molecular clusters beyond sulfuric acid systems (Zhu et al., 2014; Xu et al., 2010; Weber et al., 2012, 2014; Miao et al., 2015; Chen et al., 2017; Hu et al., 2017; Zhu et al., 2014; Odbadrakh et al., 2020; Gong et al., 2024; Chen et al., 2020). For instance, Chen et al. (2020) showed that humidity can stabilize MSA–MA clusters, significantly enhancing NPF compared to the dry system. Kildgaard et al. (2018a) developed an advanced method for identifying hydrated cluster geometries, which was later applied to
study binding strengths between water and various acids (Kildgaard et al., 2018b; Rasmussen et al., 2020).

A growing body of computational studies on atmospheric clusters increasingly supports the systematic inclusion of water in cluster modeling. To facilitate this integration, we benchmark quantum chemistry methods for their accuracy in describing hydrated clusters. Specifically, we evaluate key properties such as binding electronic energies, cluster geometries, vibrational frequencies, and binding free energies. Furthermore, we analyze the hydration distributions across different cluster sizes and
compositions and examine the cluster distribution dynamics of the most relevant systems. Thus, this work not only assesses the accuracy of current methods in describing hydrated clusters but also reveals how explicitly incorporating water can influence conclusions regarding the role of humidity in NPF.





# 2 Methods

## 2.1 Molecular system datasets

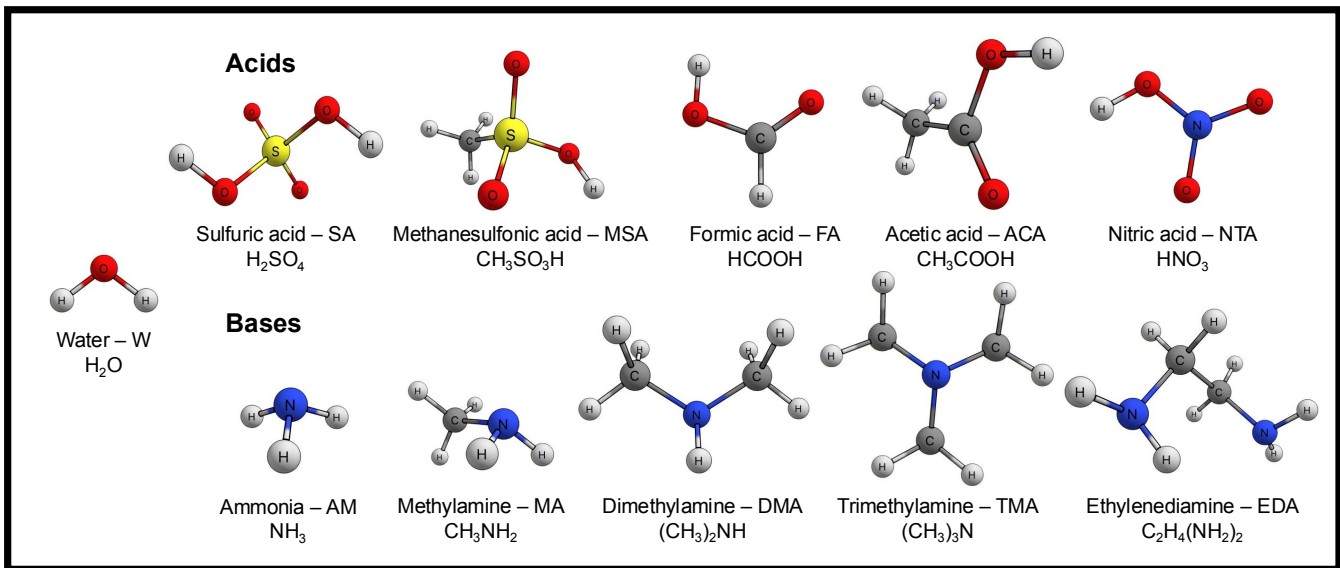

**Figure 1.** Ball-and-stick representations of the studied monomer molecules.

*Microhydrated monomer and dimer clusters* formed from various combinations of atmospherically relevant acids and bases, with varying numbers of water molecules, were used to benchmark the accuracy of the quantum chemistry (QC) methods in predicting electronic binding energies, equilibrium geometries, and cluster thermochemistry, and to investigate hydration distributions. The acids and bases included in the study are illustrated in Fig. 1. All combinations that satisfy (acid and/or base)$_{0-2}$W$_{0-5}$ were considered, resulting in a total of 395 unique clusters. For each cluster, we sampled up to five distinct

low-energy configurations ($< 50$ kcal mol$^{-1}$; see Sec. S1), optimizing their geometries at the GFN1-xTB level of theory. This resulted in a dataset of approximately 1.8k structures.

*Microhydrated (sulfuric acid–ammonia)-pair clusters*, (SA$_1$AM$_1$)$_{1-6}$W$_{0-10}$, were sampled to investigate how the electronic binding energy error and hydration distribution evolve with cluster size. For each of these cluster compositions, three unique conformers optimized at the GFN1-xTB level of theory were randomly selected from the lowest 50 kcal mol$^{-1}$ configurations

to provide a representative sampling of different cluster configurations.

*Hydrated sulfuric acid–ammonia* (SA$_{0-3}$AM$_{0-3}$W$_{0-5}$), *sulfuric acid–dimethylamine* (SA$_{0-3}$DMA$_{0-3}$W$_{0-5}$), *and methanesulfonic acid–methylamine* (MSA$_{0-3}$MA$_{0-3}$W$_{0-5}$) *clusters* were studied using cluster population dynamics to investigate the effect of humidity on the NPF rate.

A detailed description of the configurational sampling (Kubečka et al., 2023; Zhang and Dolg, 2015, 2016) procedure for

each dataset is provided in the corresponding sections and Sec. S1.




## 2.2 Benchmarked quantum chemistry methods

**Table 1.** Overview of the quantum chemistry methods and basis sets included in this benchmark study.

| Methods | | Basis sets |
|---|---|---|
| Intrinsic basis set | User-supplied basis set | |
| PM7 | M06-2X | 6-31+G(d) |
| GFN1-xTB | PW91 | 6-31++G(d,p) |
| GFN2-xTB | $\omega$B97X-D | 6-311++G(d,p) |
| AMC-xTB | RI-MP2 | 6-311++G(3df,3pd) |
| GFN1repar | DLPNO–CCSD(T$_0$) | (aug-)cc-pVDZ |
| B97-3c | DLPNO–CCSD(T$_0$)-F12 | (aug-)cc-pVTZ |
| r$^2$SCAN-3c | LNO–CCSD(T) | aug-cc-pVQZ |
| $\omega$B97X-3c | CCSD(T*)-F12 | |

We benchmarked a range of QC methods—from semi-empirical to high-accuracy wavefunction-based approaches—for their accuracy in predicting electronic binding energies and equilibrium geometries of atmospherically relevant hydrated clusters (Tab. 1). PM7 (Stewart, 2012) is a semi-empirical method based on the Hartree–Fock (HF) formalism. GFN1-xTB (Grimme et al., 2017) and GFN2-xTB (Bannwarth et al., 2019), developed by the Grimme group, are density-functional tight-binding methods. AMC-xTB (Knattrup et al., 2024) and GFN1repar (Wu et al., 2024) are reparameterizations of GFN1-xTB tailored for calculations of atmospheric molecular cluster equilibrium structures and electronic binding energies.

We also included empirically corrected DFT methods (DFT-3c) such as B97-3c (Brandenburg et al., 2018), r$^2$SCAN-3c (Grimme et al., 2021), and $\omega$B97X-3c (Müller et al., 2023), which enhance accuracy in intermolecular interactions through systematic error cancellation while maintaining computational efficiency. Additionally, we assessed hybrid and meta-generalized gradient approximation (GGA) functionals like $\omega$B97X-D (Chai and Head-Gordon, 2008) and M06-2X (Zhao and Truhlar, 2007), along with the GGA functional PW91 (Burke et al., 1998). These functionals have demonstrated reliable thermochemistry and relative binding energies for dry molecular clusters, with $\omega$B97X-D particularly noted for its consistently accurate performance (Elm and Mikkelsen, 2014; Schmitz and Elm, 2020; Jensen et al., 2022).

Last, we included the more computationally intensive wavefunction-based method RI-MP2 (Weigend et al., 1998), along with the domain-based local pair natural orbital (DLPNO; (Riplinger and Neese, 2013) and local natural orbital (LNO; Rolik et al. (2013); Nagy et al. (2018); Nagy and Kállay (2019); Kállay et al. (2020, 2025) coupled cluster methods with single, double, and perturbative triple excitations (CCSD(T)), providing robust electron correlation treatments suitable for high-precision calculations, albeit at a high computational cost.

Together, this selection offers a comprehensive range of methods, covering various levels of accuracy and computational efficiency.



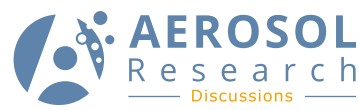

We utilized several basis sets for the QC methods that require these to be explicitly set. For M06-2X and PW91, we employed the Pople basis sets 6-31+G(d), 6-31++G(d,p), and 6-311++G(d,p) (Clark et al., 1983; Ditchfield et al., 1971; Francl et al., 1982; Gordon et al., 1982; Hariharan and Pople, 1973; Hehre et al., 1972; Spitznagel et al., 1987; Krishnan et al., 1980; McLean
and Chandler, 1980). For RI-MP2, DLPNO–CCSD($T_0$), and LNO–CCSD(T), we used the augmented correlation-consistent basis sets aug-cc-pVDZ, aug-cc-pVTZ, and aug-cc-pVQZ (Dunning, 1989; Kendall et al., 1992; Woon and Dunning, 1993). The Pople basis sets are typically more efficient, while the augmented correlation-consistent basis sets offer higher accuracy (Pitman et al., 2023). We evaluated the performance of $\omega$B97X-D using all the aforementioned basis sets, as well as the 6-311++G(3df,3pd) basis set. For LNO–CCSD(T), we employed the aug'-cc-pVTZ basis set, where the diffuse functions on
hydrogen atoms are removed. This variant is often used in noncovalent interaction and cluster studies, as diffuse functions on hydrogen typically contribute little to accuracy but can significantly increase computational cost and cause convergence issues, particularly in correlated wavefunction calculations (Del Bene, 1993). Last, the explicitly correlated (F12) technique was employed for the CCSD(T*)-F12 and DLPNO–CCSD($T_0$)-F12 methods (Pavošević et al., 2017) with the cc-pVDZ-F12 and cc-pVTZ-F12 basis set (Peterson et al., 2008).

The DLPNO–CCSD($T_0$) and LNO–CCSD(T) methods enhance efficiency compared to traditional coupled cluster methods by using a truncated set of localized or electron pair-specific natural orbitals. The choice of truncation criteria affects the number of natural orbitals included in the calculations. Tighter criteria incorporate more orbitals, improving accuracy but increasing computational cost. We evaluated the NormalPNO, TightPNO, HFC1, and HFC2 settings for DLPNO–CCSD($T_0$) and the Normal and Tight settings for LNO–CCSD(T).

QC calculations were performed with the xtb 6.7.0 (Bannwarth et al., 2021), Gaussian16 Rev.B.01 (Frisch et al., 2016), MRCC (Kállay et al., 2020, 2025), and ORCA 5.0.4 and 6.0.1 (Neese, 2012, 2022) programs. This study coincided with the release of ORCA 6.0.1, and, as a test, the B97-3c and r²SCAN-3c methods were recalculated using both versions. While the differences in calculated binding energies were negligible ($\Delta < 0.002$ kcal mol$^{-1}$), we observed an average decrease in computation time of approximately 10% for the newer version.

## 150   2.3   Electronic binding energy benchmark

Using both the *microhydrated monomer and dimer clusters* and *(sulfuric acid–ammonia)-pair clusters*, all QC methods were benchmarked based on their electronic binding energy $\Delta E_{el}$:

$$\Delta E_{el} = E_{el,cluster} - \sum_i E_{el,i}, \tag{1}$$

where $E_{el}$ is the electronic energy of the cluster/monomer, and the summation runs over all acid, base, and water molecules in
the cluster.

The quality of the benchmarked methods was assessed using signed/absolute errors and the mean absolute error (MAE), across different configurations of a cluster compared to a reference method (REF), with

$$\text{MAE} = \frac{1}{n} \sum_{i=1}^{n} |\Delta E_{el,i}^{REF} - \Delta E_{el,i}|, \tag{2}$$





where $n$ is the number of configurations, and $\Delta E_{\mathrm{el},i}^{\mathrm{REF}}$ and $\Delta E_{\mathrm{el},i}$ denote the electronic binding energies obtained with the reference method and the benchmarked QC method, respectively.

The CCSD(T*)-F12/cc-pVTZ-F12 method was used as a reference. This CCSD(T) method was chosen because it is widely regarded as the golden standard for calculating energetics (Ramabhadran and Raghavachari, 2013; Kodrycka and Patkowski, 2019), while still being computationally feasible for the small cluster sizes considered here. Kruse et al. (2020) showed that the MAE of this basis set compared to the complete basis set (CBS) limit is $0.04\,\mathrm{kcal\,mol^{-1}}$ when tested on various molecular dimers (S66 dataset; Řezáč et al. (2011)). Schmitz and Elm (2020) reported similar errors (in their SI) from CBS extrapolation for atmospheric acid–base dimers. Rescaled to our systems, we expect the reference method to have a maximum basis set incompleteness error (BSIE) of $0.1\,\mathrm{kcal\,mol^{-1}}$. Because CCSD(T*)-F12/cc-pVTZ-F12 is computationally prohibitive for all but the smallest clusters, we compared other methods to it and used the best-performing method in terms of efficiency and accuracy, DLPNO$^{\mathrm{NormalPNO}}$-CCSD(T$_0$)/aug-cc-pVTZ (see Secs. 3.1 and 3.2), as the reference for larger clusters.

All QC output is stored in the Atmospheric Cluster Database (ACDB; Elm and Kubečka (2024); Elm (2019); Kubečka et al. (2023); see Sec. S2).

### 2.4 Equilibrium geometry benchmark

To evaluate how well different efficient QC methods approximate equilibrium geometries of hydrated clusters and to assess the correlations between them, we compared up to five configurations across methods for the (acid and/or base)$_{0-2}$W$_{0-5}$ clusters. All sampled equilibrium geometries obtained at $\omega$B97X-D/6-31++G(d,p) level of theory (see Sec. S1) were further reoptimized with each benchmarked method. The ArbAlign program was used to align the optimized geometries and calculate root-mean-square deviations (RMSD) between them (Temelso et al., 2017). The methods were compared with each other to identify inter-method correlations and against the RI-MP2/aug-cc-pVQZ reference, which is known to provide accurate geometries compared to higher levels of theory (e.g., DF-CCSD(T*)-F12b/cc-pVDZ-F12) (Jensen and Elm, 2024; Coriani et al., 2005).

### 2.5 Thermochemistry benchmark

QC combined with statistical thermodynamics enables the calculation of thermochemical properties of molecular clusters, such as their Gibbs free energies of formation. The accuracy of such data is difficult to assess due to little experimental data being available. In Sec. 3.3, we place particular emphasis on vibrational frequencies, which are used to construct the vibrational partition function for Gibbs free energy calculations. The ability of a QC method to produce accurate vibrational frequencies therefore serves as an indicator of its reliability in predicting thermochemical data. In addition, we investigate other potential sources of error in computed thermochemical properties and evaluate their possible magnitudes.

### 2.6 Hydration distributions

We examined hydration distributions for the (acid and/or base)$_{0-2}$W$_{0-5}$ systems, as well as hydrated clusters of SA–AM, at 278.15 K and 298.15 K. The QC methods that performed best in terms of efficiency and accuracy, based on our benchmarks,



were used for these calculations. The population $x_n$ of a cluster containing $n$ water molecules is given by (Henschel et al., 2014)

$$x_n = \left( \frac{p_{\mathrm{H_2O}}}{p^0} \right)^n x_0 \cdot e^{-\Delta_{\mathrm{hydr}} G_n / k_{\mathrm{B}} T}, \tag{3}$$

where the population of the dry cluster, $x_0$, is chosen such that $\sum_{n=1} x_n = 1$, with the sum extending to the most hydrated cluster considered. Here, $p_{\mathrm{H_2O}}$ is the water partial pressure, $p^0$ the reference pressure (1 atm), and $\Delta_{\mathrm{hydr}} G_n$ the standard Gibbs free energy of hydration, i.e. $\Delta_{\mathrm{hydr}} G_n = \Delta G_n - \Delta G_0$. Relative humidity (RH) is calculated with respect to saturation vapor pressure ($p^0_{\mathrm{H_2O}}$) as RH $= p_{\mathrm{H_2O}} / p^0_{\mathrm{H_2O}} \cdot 100\%$, while we obtained $p^0_{\mathrm{H_2O}}$ using the August–Roche–Magnus equation (Alduchov and Eskridge, 1996; Westermann et al., 2016).

### 2.7 Particle formation rate calculations

To investigate the impact of hydration on NPF, we calculated particle formation rates ($J$) for the $\mathrm{SA}_{0-3}\mathrm{AM}_{0-3}\mathrm{W}_{0-5}$, $\mathrm{SA}_{0-3}\mathrm{DMA}_{0-3}\mathrm{W}_{0-5}$, and $\mathrm{MSA}_{0-3}\mathrm{MA}_{0-3}\mathrm{W}_{0-5}$ systems using the Atmospheric Cluster Dynamics Code (ACDC) (McGrath et al., 2012; Olenius et al., 2013; Olenius, 2018). Cluster evaporation rates were derived from the binding Gibbs free energies of the clusters, calculated using the most reliable methods identified in the electronic energy and equilibrium geometry benchmarks.

$J$ was evaluated over a 0–100% relative humidity range at temperatures of 278.15 K and 298.15 K. The following constant monomer concentrations were used: SA and MSA at $10^5$ and $10^7$ cm$^{-3}$; AM at 10 and 10,000 ppt; DMA at 1 and 10 ppt; and MA at 1 and 10 ppt, covering typical boundary-layer ranges.

Coagulation loss (CL) of clusters was included using CL $= 10^{-3}(d/d_{\mathrm{SA}})^{-1.6}$, where $d$ is the cluster diameter and $d_{\mathrm{SA}}$ that of the SA monomer (Maso et al., 2008). Clusters larger than (acid)$_3$(base)$_3$ were considered as spontaneously outgrowing into particles. For instance, acid, base, and cluster addition to the hydrated (acid)$_3$(base)$_3$ clusters were allowed to grow out of the simulations and contribute to the particle formation rate. A more detailed description of the ACDC simulations is provided in Sec. S3.

## 3 Results

### 3.1 Electronic binding energy benchmark

#### 3.1.1 Hydrated monomers and dimers

Single point calculations at the CCSD(T*)-F12/cc-pVTZ-F12 reference method were feasible for only ∼400 small clusters within a reasonable computation time on our hardware. To assess alternative reference methods, we first compared high-quality RI-MP2, DLPNO, and LNO approaches against CCSD(T*)-F12/cc-pVTZ-F12 for this subset. The top part of Fig. 2 presents violin plots of the absolute errors alongside the average CPU time per single-point energy calculation. Calculations were run on either Intel Xeon Gold 6248R or Intel Xeon Platinum 8358 CPUs. Since CPU time is hardware-dependent, the reported times





**Figure 2.** Violin plots of absolute errors in electronic binding energies for microhydrated monomers and dimers. Height indicates error magnitude, width represents the number of configurations with the same error, and vertical black lines mark mean absolute errors. The dataset includes all 395 unique combinations of the molecules in Fig. 1, with formula (acid and/or base)$_{0-2}$(water)$_{0-5}$. Top: RI-MP2, DLPNO, and LNO methods, benchmarked against CCSD(T*)-F12/cc-pVTZ-F12 for a subset of ∼0.4k small clusters. Bottom: Semi-empirical, DFT-3c, and $\omega$B97X-D methods compared to DLPNO$^{\text{NormalPNO}}$-CCSD(T$_0$)/aug-cc-pVTZ for the full data set of ∼1.8k clusters. Average CPU times are given in seconds ($''$), minutes ($'$), or hours (h).





are only indicative. Several tested methods closely agree with CCSD(T*)-F12/cc-pVTZ-F12. Among them, DLPNO$^{\text{NormalPNO}}$-CCSD(T$_0$)/aug-cc-pVTZ stands out with a low mean absolute error (MAE) of $\sim$0.20 kcal mol$^{-1}$, low memory requirements, and an average computational cost under 1 CPU hour. This is consistent with the findings of Schmitz and Elm (2020). We therefore selected it as the reference method for the full dataset of $\sim$1.8k structures. However, we also highlight the accuracy of the LNO methods, along with their low memory requirements, which stem from the use of local MP2 natural orbitals. In contrast, DLPNO requires computing and storing pair natural orbitals. For example, single-point calculations on SA$_1$TMA$_1$W$_5$ demonstrate that LNO$^{\text{Tight}}$-CCSD(T)/aug-cc-pVTZ requires only 2 GB of memory, whereas DLPNO$^{\text{NormalPNO}}$-CCSD(T$_0$)/aug-cc-pVTZ requires 6 GB. We chose the DLPNO method over the LNO methods for consistency with previous studies rather than for their accuracy difference.

In the full dataset comparison, the semi-empirical methods are extremely fast, with mean CPU times of 0.4–3 seconds. Among them, GFN1-xTB and GFN2-xTB exhibit the lowest MAEs, with GFN1-xTB showing a slightly lower maximum in absolute electronic binding energy error. For many dry clusters, GFN1-xTB has been found to outperform GFN2-xTB as well (Jensen et al., 2022; Rasmussen et al., 2022; Wu et al., 2023; Engsvang and Elm, 2022). However, GFN2-xTB is significantly faster, requiring only 0.5 seconds per calculation compared to 3 seconds for GFN1-xTB. AMC-xTB, a reparameterization of GFN1-xTB for dry molecular clusters, performs worse than the original GFN1-xTB for these microhydrated monomers and dimers, likely because no water-containing clusters were included during the reparameterization.

The r$^2$SCAN-3c and $\omega$B97X-3c methods perform particularly well, with MAEs of 1.29 and 1.26 kcal mol$^{-1}$ but still with maximum absolute errors of $\sim$6.5 and $\sim$6.1 kcal mol$^{-1}$, respectively. However, $\omega$B97X-3c requires more than three times the CPU time of the other two DFT-3c methods studied.

In Sec. S4, we show signed electronic binding energy errors for all benchmarked QC methods. Among the M06-2X, PW91, and $\omega$B97X-D functionals, $\omega$B97X-D performs best when using the same basis set. Paired with the aug-cc-pVQZ basis set, $\omega$B97X-D achieves an MAE of 1.25 kcal mol$^{-1}$, comparable to the 1.26 kcal mol$^{-1}$ of $\omega$B97X-3c, though at a significantly higher computational cost of 35 CPU hours compared to just 7 CPU minutes for $\omega$B97X-3c. In contrast, the commonly used combination of $\omega$B97X-D with the 6-31++G(d,p) basis set has a similar CPU time of 7 CPU minutes to $\omega$B97X-3c but results in a significantly higher MAE of 6.07 kcal mol$^{-1}$. For comparison, the DLPNO$^{\text{NormalPNO}}$-CCSD(T$_0$)/aug-cc-pVTZ reference method required an average of 5.3 CPU hours for the full dataset.

Based on this electronic binding energy benchmark, the r$^2$SCAN-3c and $\omega$B97X-3c methods stand out as excellent choices for fast and accurate binding energy calculations of hydrated acid–base clusters.

### 3.1.2 Scaling with cluster size

To examine how the accuracy of electronic binding energies from the QC methods evolves with increasing cluster size, we analyzed all clusters satisfying the composition (SA$_1$AM$_1$)$_{1-6}$W$_{0-10}$. Only a subset of the best-performing QC methods—those offering a balance of efficiency and accuracy based on the electronic binding energy benchmark in the previous section—was included. The DLPNO$^{\text{NormalPNO}}$-CCSD(T$_0$)/aug-cc-pVTZ method was again used as the reference, as it remains computationally feasible even for relatively large clusters.



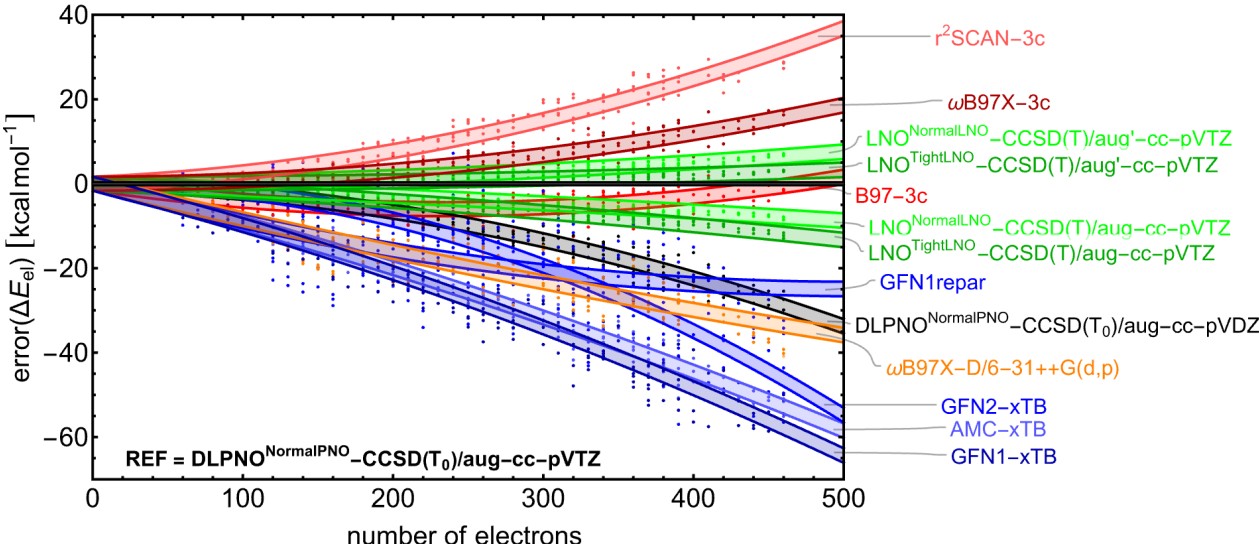

**Figure 3.** The signed error in (electronic) binding energy ($kcal\,mol^{-1}$) relative to the DLPNO$^{NormalPNO}$-CCSD(T$_0$)/aug-cc-pVTZ reference calculations as a function of the number of electrons for multiple quantum chemistry methods. Electronic energies were determined from single-point energy calculations for three conformers across all clusters with the composition $(SA_1AM_1)_{1-6}W_{0-10}$, randomly selected from the GFN1-xTB optimized geometries. A second-order polynomial, constrained to pass through the origin [0,0], was fit to the data for each quantum chemistry method. The shaded area illustrates the 90% confidence intervals for individual predictions based on this fit.

Figure 3 shows the signed error in electronic binding energy as a function of the number of electrons in the system for
each of the included QC methods relative to the reference. For visualization, a second-order polynomial passing through the origin was fitted to the data for each QC method, with the shaded area representing the 90% confidence interval for single predictions. As expected, the semi-empirical xTB methods show the largest errors, with values reaching $-60\ kcal\,mol^{-1}$ for the largest clusters. AMC-xTB, a reparameterization of GFN1-xTB tailored to reproduce $\omega$B97X-D/6-31++G(d,p) data for dry atmospheric clusters, performs slightly better than GFN1-xTB. For this benchmark, we also included the GFN1repar
method, a reparameterization of GFN1-xTB aimed at reproducing B97-3c data for large dry atmospheric clusters (fitted up to $(acid)_{10}(base)_{10}$ clusters). GFN1repar shows significant improvement in accuracy as cluster size increases, outperforming the other semi-empirical methods. While the other semi-empirical methods exhibit a more than linear increase in error magnitude with cluster size, GFN1repar demonstrates a slightly less-than-linear trend, which can be attributed to its focus on large clusters and its fitting to B97-3c, a method that performs better but exhibits similar behavior.

The performance of DLPNO$^{NormalPNO}$-CCSD(T$_0$) with a double zeta basis set highlights the significance of basis set size, as an error of approximately $-30\ kcal\,mol^{-1}$ in electronic binding energy is observed for the largest clusters studied here. The low errors associated with LNO–CCSD(T) methods (even lower with tighter LNO criteria and reduced diffuse functions for hydrogens) validate the choice of the reference method used in this study. Given its high memory efficiency compared to





DLPNO, LNO methods are recommended for calculating the properties of large clusters in future studies (Knattrup and Elm, 2025).

The DFT-3c methods B97-3c and $\omega$B97X-3c perform exceptionally well, especially considering their efficiency. Although the error of $\omega$B97X-3c increases with the number of electrons, it does so less steeply than r$^2$SCAN-3c and $\omega$B97X-D/6-31++G(d,p), resulting in absolute errors lower than 20 kcal mol$^{-1}$ for the largest cluster studied here. B97-3c shows an increasingly negative error up to $\sim$250 electrons, after which the error magnitude decreases, resulting in a small positive error around 500 electrons. The absolute errors stay below 10 kcal mol$^{-1}$ for all studied cluster sizes, with a remarkably low error for the largest cluster due to the aforementioned trend.

For the SA–AM–W clusters studied here, the DFT-3c methods outperform all other fast methods regarding the accuracy of electronic binding energies relative to the chosen reference method. Given its exceptional performance in both the cluster size benchmark and the binding energy analysis of microhydrated monomers and dimers in the previous section, along with being approximately 1 to 2 orders of magnitude faster than the reference, $\omega$B97X-3 could serve as an efficient method for obtaining reasonable electronic binding energies for large hydrated atmospheric clusters. This conclusion aligns with recent findings for small dry clusters reported by Jensen and Elm (2024).

## 3.2 Equilibrium Cluster Geometry

Figure 4 shows the mean RMSD for selected methods, while results for the other methods are provided in Sec. S5. All $\sim$1.8k optimization were performed with default setting, except for the RI-MP2/aug-cc-pVQZ reference, where only $\sim$0.6k converged with extreme SCF and very tight optimization criteria. Compared to the reference, both $\omega$B97X-D/6-31++G(d,p) and $\omega$B97X-3c perform best with an RSMD of 0.04 Å. Besides the DFT-3c methods, $\omega$B97X-D/6-31++G(d,p) clearly performs the best, as has also been shown in previous studies (Jensen and Elm, 2024). However, $\omega$B97X-3c yields a similar accuracy, alongside accurate electronic binding energy benchmarks presented in Section 3.1. r$^2$SCAN-3c also shows good agreement with the RI-MP2/aug-cc-pVQZ reference. Given that r$^2$SCAN-3c is significantly more computationally efficient than $\omega$B97X-D/6-31++G(d,p) and $\omega$B97X-3c, it is well-suited for use as an intermediate optimization method during configurational sampling of hydrated clusters. Interestingly, GFN2-xTB performs better than GFN1-xTB. The opposite conclusion was reported for dry atmospheric clusters (Jensen and Elm, 2024). While less accurate, the xTB methods are fast and thus suitable for geometry pre-optimization. The PM7 method seems less suitable for molecular clusters.

## 3.3 Cluster Thermochemistry

The thermodynamics of molecular systems arise from their dynamics on the potential energy surface (PES), where they transition between discrete vibrational–rotational–translational energy levels associated with different configurational minima. Although the previous sections indicated a low MAE across QC methods for electronic binding energies and equilibrium geometries at the GFN1-xTB level—suggesting that the PES shape is generally well reproduced—this does not necessarily guarantee that thermochemical properties are accurately captured.



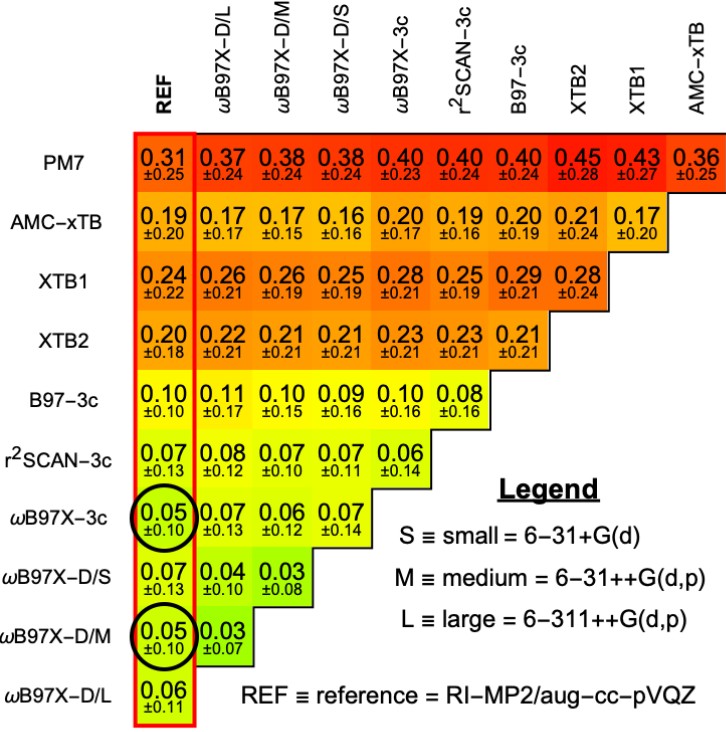

**Figure 4.** Mean of root-mean-square deviations (RMSD [Å]) between (acid and/or base)$_{0-2}$W$_{0-5}$ equilibrium geometries optimized at different levels of theory. Black circles indicate the best-performing methods relative to the reference, RI-MP2/aug-cc-pVQZ (REF). More detailed comparison is presented in Sec. S5.

Benchmarking thermochemical properties of atmospheric molecular clusters is particularly challenging due to the scarcity of reliable reference data, and is therefore often omitted in methodological evaluations. In this section, we explicitly address thermochemical aspects of hydrated molecular clusters. The dominant contributions arise from vibrations around the most populated minimum-energy conformer. To improve accuracy beyond the harmonic oscillator approximation, we consider standard

corrections, including anharmonicity scaling of vibrational frequencies, quasi-harmonic treatment of low-frequency modes, and multi-conformer Boltzmann averaging to account for transitions between multiple low-energy conformers. For comparison, we will also derive thermochemical corrections from umbrella sampling, as demonstrated by Kubečka et al. (2025).

Figure 5 shows the free energy contributions from these corrections as a function of the number of atoms in the cluster. Halonen (2024) used MD simulations with force-field methods to demonstrate that the combined effects of anharmonicity and

interconversion between minima can reach up to $k_BT/4$ per vibrational mode. We refer to this upper bound as the Halonen thermodynamic limit (black line).





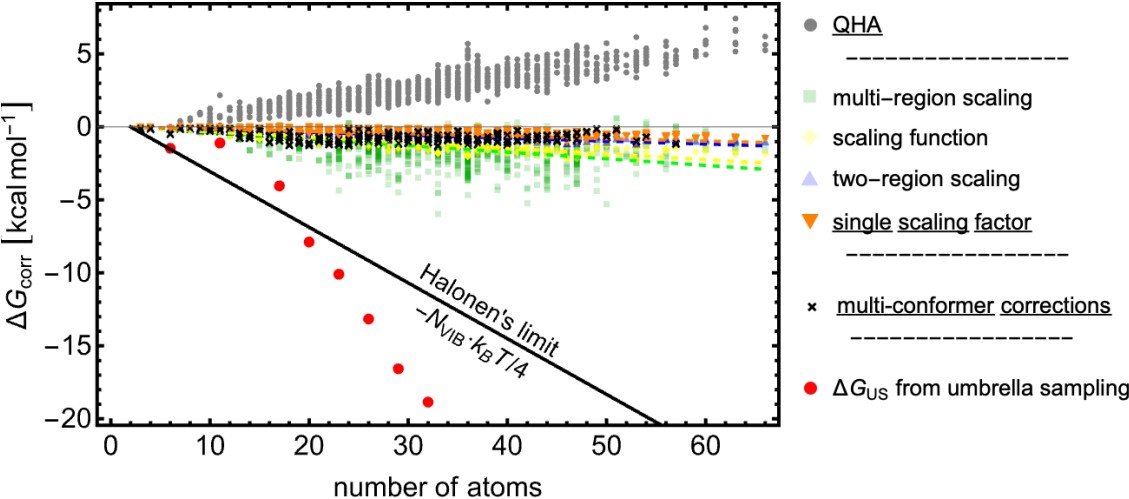

**Figure 5.** Free energy contributions from vibrational corrections (colored points corresponding to different scalings) and from accounting for multiple low-lying free energy minima, calculated using Eq. 4 as derived by Partanen et al. (2016) (black circles).

Each correction is discussed in detail in the following subsections. While a full quantitative treatment of anharmonicity is beyond the scope of this study, discussing these corrections and estimating their expected magnitudes provides a more physically realistic description of the vibrational and conformational contributions to the free energies of hydrated molecular clusters.

### 3.3.1 Low-vibrational frequency treatment

Low-vibrational frequencies are common in hydrated molecular clusters and may even play an important role in their stabilization. However, within the harmonic oscillator treatment and due to numerical inaccuracies, such frequencies can be underestimated, leading to too low values. This poses a problem because the entropic contribution diverges to $-\infty$ as the vibrational frequency approaches zero. To mitigate this issue, the quasi-harmonic approximation (QHA; (Grimme, 2012)) replaces the vibrational entropy of low-frequency modes with the corresponding rotational entropy. In this work, we applied a smooth rotor–vibration transition with crossover frequency of $100\,\mathrm{cm^{-1}}$. As shown in Fig. 5, the QHA correction to the Gibbs free energy is generally positive and increases approximately linearly with the number of atoms, reaching $\sim 5\,\mathrm{kcal\,mol^{-1}}$ for a 60-atom cluster. It is worth noting that some programs (e.g., ORCA and XTB) already apply this approximation by default.

### 3.3.2 Vibrational anharmonicity correction

Vibrational frequencies are typically calculated within the rigid-rotor harmonic oscillator (RRHO) approximation, which neglects anharmonicity. This harmonic treatment is a significant source of error when comparing calculated vibrational frequencies to experimental results. Halonen (2024) highlighted that anharmonicity becomes increasingly important for larger clusters,





**Table 2.** The single scaling factor obtained by fitting to the difference between the anharmonic (VPT2) and harmonic vibrational frequencies compared to list of experimental observations. See details in Sec. S6.

|  | B97-3c | r²SCAN-3c | ωB97X-3c | ωB97X-D/6-31++G(d,p) | RI-MP2/aug-cc-pVQZ |
|---|---|---|---|---|---|
| no scaling (scaling factor) | <1> | <1> | <1> | <1> | <1> |
| MAE [cm⁻¹] | 61 | 86 | 131 | 110 | 97 |
| single scaling factor | <0.944> | <0.950> | <0.954> | <0.950> | <0.95 (Johnson, 1999)> |
| MAE [cm⁻¹] | 81 | 52 | 60 | **39** | 40 |

as the number of vibrational modes grows with cluster size. The harmonic approximation generally overestimates vibrational frequencies by 2–6% (Lin et al., 2008).

Second-order vibrational perturbation theory (VPT2; Joel M Bowman and Meyer (2008)), as implemented in ORCA (Barone et al., 2014), can be used to account for anharmonicity. VPT2 typically reproduces experimental fundamental vibration frequencies to within 0–30 cm⁻¹. However, due to practical limitations such as computational cost and convergence issues, vibrational scaling factors are often applied as a simpler alternative. Jacobsen et al. (2013) further note that anharmonic vibrational frequency calculations are not worthwhile when small basis sets are used, as the error introduced by the basis set exceeds that from neglecting anharmonicity, making scaling factors sufficient.

Scaling factors have been defined for various methods (Johnson, 1999; Myllys et al., 2016). However, we fitted our own scaling corrections for the (acid and/or base)$_{0-2}$W$_{0-5}$ clusters, testing four approaches: (1) a single constant scaling factor across the full frequency range, (2) two separate constant scaling factors for the regions below and above 2000 cm⁻¹, (3) multi-region constant scaling factors with a region size of 100 cm⁻¹, and (4) a flexible scaling function of the form $A - B \cdot \nu_{\mathrm{harm}} - C/(D + \nu_{\mathrm{harm}})$. The details of the fitting and analysis are provided in Sec. S6. Here, we only summarize that we find that a single scaling factor is sufficient, with no significant improvement from more complex corrections. Table 2 shows the MAEs relative to experimental (Dunn et al., 2006; Huber and Herzberg, 1979; Shimanouchi et al., 1972; Otto et al., 2014; Rognoni et al., 2021; Vogt and Kjaergaard, 2022; Hintze et al., 2003; Rozenberg et al., 2012; Kjaersgaard et al., 2020; Soulard and Tremblay, 2021; Fateley and Miller, 1962; Li et al., 2016; Fernández et al., 2005; Herzberg, 1966; Koops et al., 1983; Zhang et al., 2021; Telfah et al., 2024; Lewandowski et al., 2005; Maroń et al., 2009; McCurdy et al., 2002) vibrational frequencies with and without applying a single scaling factor. For r²SCAN-3c, ωB97X-3c, ωB97X-D/6-31++G(d,p), and RI-MP2/aug-cc-pVQZ scaling significantly improves the MAE compared to the unscaled harmonic approximation. In contrast, B97-3c yields the lowest MAE without scaling, and applying scaling functions actually worsens agreement with experiment.

Figure 5 shows the anharmonicity correction to the binding free energy of all studied microhydrated monomer, dimer, and (sulfuric acid–ammonia)-pair clusters. The magnitude of the vibrational corrections decrease with increasing system size. With the exception of the multi-region scaling, all scaling approaches reduce the cluster Gibbs free energies to a similar extent, indicating that a single scaling factor provides a reasonable first approximation. In general, applying a single scaling factor systematically lowers reaction and addition free energies, as the Gibbs free energy correction scales approximately

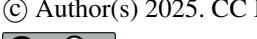



linearly with system size. The multi-region scaling correction is significantly larger, which could in principle account for missing contributions to the binding free energies of large clusters. However, when tested for $\omega$B97X-3c, the correction was not significantly more accurate than that obtained with a single scaling factor. The multi-region scaling is thus very sensitive to the data. We also reiterate that this scaling performs worse compared to experimental data.

      For comparison, Temelso et al. (2011) studied $W_{1-10}$ clusters at the CCSD(T)/CBS//RI-MP2/aVDZ level of theory and

found that the anharmonic correction scales linearly with the number of atoms, amounting to $-4.1\,\mathrm{kcal\,mol^{-1}}$ for the 30-atom $W_{10}$ cluster. This suggests that anharmonic effects in some systems or for some methods could be even more significant than those reported here. A more comprehensive treatment of anharmonicity in molecular clusters would therefore be valuable. But, this is theoretically challenging because the multidimensional PES of hydrogen-bonded clusters contains many shallow minima, and coordinate choices strongly influence how anharmonic couplings are represented. Hence, such an analysis lies

beyond the scope of the present benchmarking study.

### 3.3.3   Multi-conformational contribution

In QC, properties are commonly calculated for the lowest free energy conformation, under the assumption that no other low-energy conformers are significantly populated. To assess the importance of multi-conformational entropy, we used the analytical expression of Partanen et al. (2016). According to Partanen et al. (2016), the multi-conformer binding free energy correction

$\Delta G_{\mathrm{mc}}$ is given by:

$$\Delta G_{\mathrm{mc}} = -k_{\mathrm{B}}T\ln\sum_{i} e^{-\Delta\Delta G_i/k_{\mathrm{B}}T}, \tag{4}$$

where $\Delta\Delta G_i$ denotes the Gibbs binding free energy difference of conformer $i$ relative to the lowest free-energy structure, $k_{\mathrm{B}}$ is the Boltzmann constant, and $T$ the temperature. The summation runs over all unique conformers.

      Large hydrated clusters can posses multiple significantly populated free energy minima. We, therefore, calculated the multi-

conformational free energy contributions using Eq. 4 at $\omega$B97X-D/6-31++G(d,p) for all clusters fulfilling $SA_{0-3}AM_{0-3}W_{0-8}$. As shown in Fig. 5, the resulting corrections are relatively small, never exceeding $-2\,\mathrm{kcal\,mol^{-1}}$. Moreover, these results do not increase with the number of atoms, indicating that multi-conformational contributions do not become more significant with cluster size. This observation is consistent with the findings of Halonen (2024), who noted that large clusters predominantly occupy a single low-lying minimum.

### 3.3.4   Umbrella Sampling simulations

In a recent study (Kubečka et al., 2025), we performed umbrella sampling simulations (Torrie and Valleau, 1974) with the PaiNN machine learning potential (Schütt et al., 2019, 2023; Schütt et al., 2021) to calculate cluster Gibbs binding free energies through an approach independent of the statistical thermodynamics traditionally applied in combination with QC results. This approach inherently accounts for anharmonic effects and multi-conformer contributions. In that work, we com-

pared reaction Gibbs free energies obtained from *umbrella sampling* with those from the *traditional QC approach combined with the quasi-harmonic approximation (QHA)* for the $W_1+W_1$ and $SA_1+AM_1$ systems. Here, we extend this analysis to the

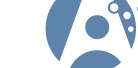

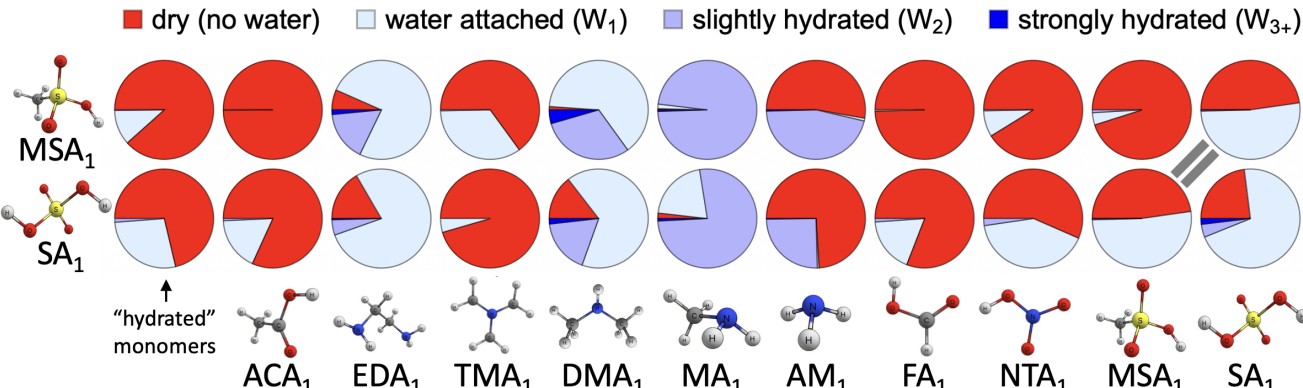

**Figure 6.** Pie charts of the hydration distribution at 100% relative humidity and 278.15 K for all studied monomers and dimers with up to five water molecules. No hydration is indicated in red, while clusters with one to five water molecules are represented with increasingly darker shades of blue. The results were obtained for the lowest free energy minimum at the DLPNO//DFT level of theory using quasi-harmonic approximation.

$SA_1DMA_1W_n+W_1$ clusters ($n = 0$–$4$), following the same computational protocol. For additional methodological details, we refer the reader to our earlier work (Kubečka et al., 2025). As shown in Fig 5, the combined application of QHA (gray) and US (red) corrections yields values close to the Halonen limit, suggesting that the traditional QC approach may significantly underestimate cluster binding free energies, and the anharmonic corrections indeed might be close to the Halonen limit. This methodology was not yet verified but shows that there might be some missing entropic effects in the traditionally applied statistical thermodynamics. In the following sections, we will therefore consider both QC Gibbs binding free energies with QHA corrections, and those additionally corrected according to the Halonen limit ($-k_BT/4$ per vibrational mode). This provides a practical lower and upper bound for the expected binding free energies.

## 3.4 Hydration distribution

Based on the benchmarking of electronic energies, equilibrium geometries, and thermochemical properties in the previous sections, we chose DLPNO$^{NormalPNO}$–CCSD(T$_0$)/aug-cc-pVTZ//$\omega$B97X-D/6-31++G(d,p), hereafter abreviated as DLPNO//DFT, as the method for calculating hydration distributions. This choice was motivated by the good performance of the DFT method in the equilibrium structure and vibrational analysis benchmarks, while DLPNO excelled in the electronic energy benchmark. Note that this level of theory was already recommended as suitable for molecular clusters in previous studies (Elm et al., 2020; Smith et al., 2021; Trolle et al., 2025). Here, we examine the hydration distributions of various acid–base clusters. First, we consider only the lowest free energy minimum corrected by QHA. Under atmospheric conditions, most monomers and dimers remain predominantly unhydrated (see Sec. S7). Only a few dimers containing sulfuric acid (SA) and methanesulfonic acid (MSA) are more likely to be hydrated than not, as illustrated in Fig. 6. In the figure, unhydrated clusters (i.e., with zero water molecules attached) are shown in red, while clusters with one to five water molecules are represented by increasingly darker





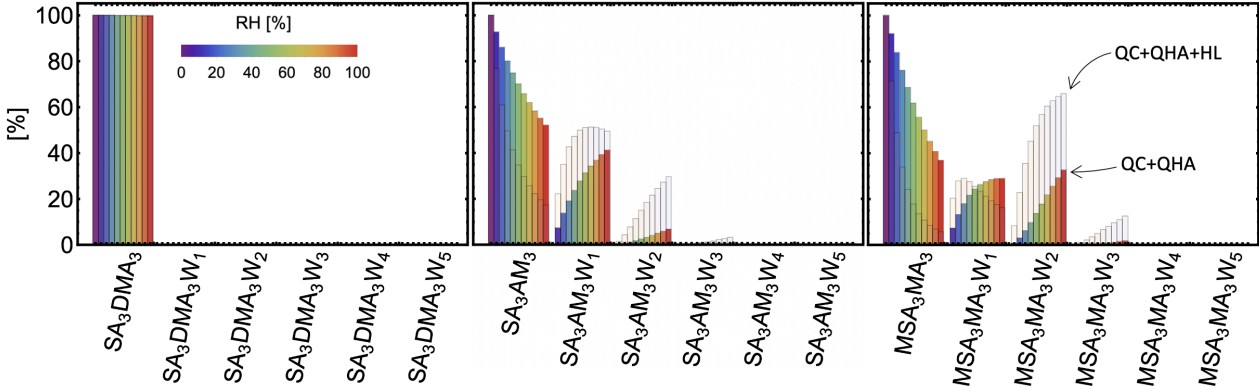

**Figure 7.** Hydration distributions of the $SA_3AM_3$, $SA_3DMA_3$, and $MSA_3MA_3$ clusters at 278.15 K. The humidity level is indicated by different color. The results were obtained for the lowest free energy minimum at the DLPNO//DFT level of theory using quasi-harmonic approximation. The transparent distribution shows the change due to the free energy correction at the Halonen limit.

shades of blue. The resulting hydration distribution varies with acidity/basicity and structural factors, such as steric hindrance and the availability of hydrogen-bonding sites. Under the same conditions, we also examined the hydration distributions of $SA_3AM_3$, $SA_3DMA_3$, and $MSA_3MA_3$. Figure 7 reveals that the $SA_3DMA_3$ remains completely dry, which we attribute to the steric effects of the methyl groups. $MSA_3MA_3$ is the most hydrated among the three, yet it is still less hydrated than

410 $MSA_1MA_1$, again likely due to the presence of the methyl groups. In contrast, $SA_3AM_3$ shows increased hydration compared to $SA_1AM_1$.

The hydration of the SA–AM and SA–DMA systems has previously been studied by Myllys et al. (2021) and Henschel et al. (2014, 2016), and our hydration distributions from QHA-corrected single-structure cluster thermochemistry correspond quite well to their results. Chen et al. (2020) further suggested that humidity strongly enhances the NPF of the MSA–MA system,

by enabling stabilizing proton transfers. We critically revisit this NPF enhancement in Sec. 3.5. While our results confirm that MSA–MA is more hydrophilic, the effect appears less pronounced than reported by Chen et al. (2020). Additionally, Ge et al. (2020) found that $TMA_1$ is hydrophobic but, in contrast to our results, reported that $DMA_1$ is almost always hydrated with one water molecule. We, therefore, examine potential sources of this discrepancy. Rather than temperature dependence (within the atmospheric window), differences in the chosen quantum chemistry methods and thermochemical corrections appear to play a

decisive role in shaping the predicted hydration distributions.

As shown in Fig 7, applying the full Halonen-limit correction to the thermochemistry enhances cluster hydration, although the effect remains moderate (cf. the transparent histogram). To examine how the hydration distribution depends on the choice of QC method, both with and without Halonen limit corrections, we calculated the binding free energy of all $(SA_1AM_1)_{1–3}W_{0–5}$ clusters using two different methods: $\omega B97X$-3c and DLPNO//DFT. Additionally, we calculated the hydration distribution for

SA–AM clusters up to $SA_6AM_6$ with up to ten water molecules using the more efficient B97-3c method. We also present the electronically corrected composite methods: LNO//B97-3c, LNO//$\omega B97X$-3c, and DLPNO//$\omega B97X$-3c. Here we omitted

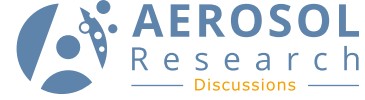

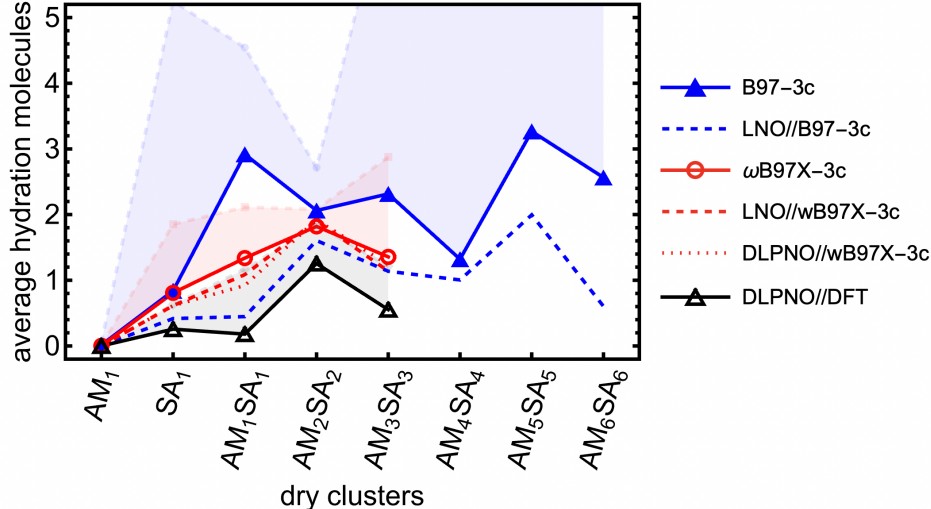

**Figure 8.** Average hydration of sulfuric acid (SA)–ammonia (AM) clusters calculated at 100% relative humidity and 278.15 K. The colors represents which method was used for geometry optimization and thermochemistry (B97-3c, $\omega$B97X-3c, and DFT), while some of the methods are corrected with single-point calculation at $\text{DLPNO}^{\text{NormalPNO}}$–CCSD($T_0$)/aug-cc-pVTZ and $\text{LNO}^{\text{Tight}}$–CCSD(T)/aug'-cc-pVTZ level of theory. Bright lines are for calculations with one-structure and quasi-harmonic approximations. The low-opacity line-points corresponds to systematically applied Halonen limit thermodynamics correction to all cluster binding free energies.

DLPNO//B97-3c as it would be slow for the largest clusters. The resulting average numbers of attached water molecules at 278.15 K are presented in Fig. 8. Overall, B97-3c predicts more hydration for $SA_1AM_1$, $SA_2AM_2$, and $SA_3AM_3$ than $\omega$B97X-3c, while $\omega$B97X-3c in turn predicts more hydration than DLPNO//DFT. However, the relative trends are not consistent across cluster sizes. For example, B97-3c predicts reduced hydration when moving from $SA_1AM_1$ to $SA_2AM_2$, while DLPNO//DFT predicts an increase in hydration. However, after applying electronic energy corrections to the B97-3c method we observe similar trends. LNO//B97-3c could potentially be a new emerging method for fast calculations. Moreover, similar results are obtained for $\omega$B97X-3c, LNO//$\omega$B97X-3c, and DLPNO//$\omega$B97X-3c, which shows that $\omega$B97X-3c could emerge as a new reasonably accurate method for large clusters without a need for electronic correction. Nevertheless, there are still some discrepancies in hydration distributions between methods, and thus relative comparisons should only be made within the same method. Applying the systematic Halonen limit correction further increases the predicted hydration, with the effect being most pronounced for methods that yield stronger binding energies.

When examining the evolution of the hydration distribution with cluster size using the B97-3c method, we find that clusters with even numbers of SA and AM molecules (i.e., $SA_2AM_2$, $SA_4AM_4$, $SA_6AM_6$) are less hydrated than the neighboring clusters with odd numbers of SA and AM molecules. The dry structures with even SA and AM are highly symmetric, resulting in lower energies than the trend in binding free energies with cluster size would suggest. Interestingly, this pattern is not observed with the other two methods. Overall, hydration is rather minor but seems to slowly increase with cluster size because



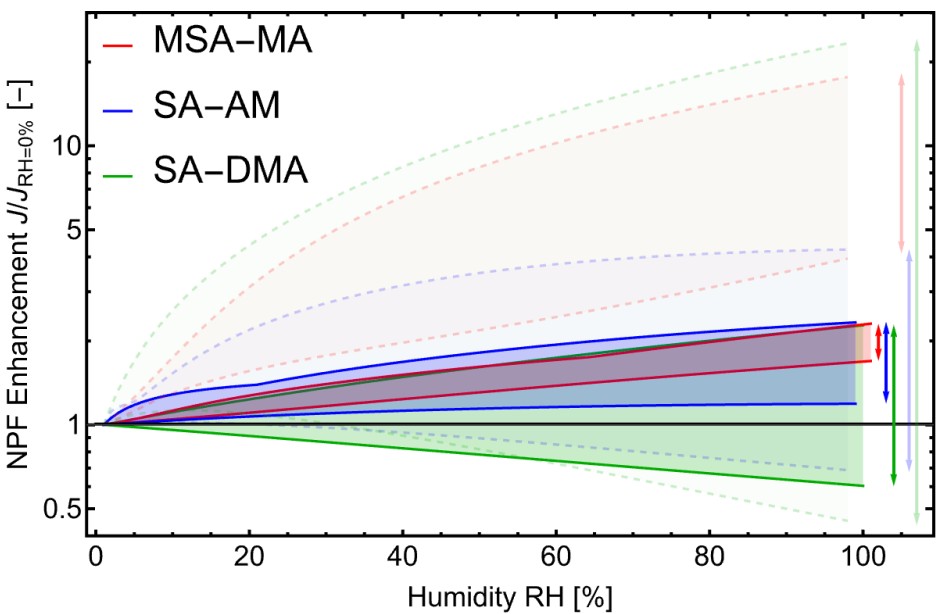

**Figure 9.** The enhancement in particle formation rate, $J/J_{\mathrm{RH=0\%}}$, as a function of relative humidity (RH), calculated with the Atmospheric Cluster Dynamics Code (ACDC) for clusters up to $\mathrm{acid_3 base_3 water_5}$. Results are shown for the methanesulfonic acid–methylamine (MSA–MA; red), sulfuric acid–ammonia (SA–AM; blue), and sulfuric acid–dimethylamine (SA–DMA; green) systems. Solid lines correspond to DLPNO//DFT quantum-chemical data with QHA, while dashed, semi-transparent lines show results corrected to the Halonen limit. Simulations were performed at 278.15 and 298.15 K, with sulfuric acid concentrations of $10^5$ and $10^7$ cm$^{-3}$, methylamine concentrations of 1 and 100 ppt, ammonia concentrations of 10 and 10,000 ppt, and dimethylamine concentrations of 1 and 10 ppt.

larger clusters can accommodate more water molecules, either by incorporating them into acid–base interaction bridges or by exposing greater surface area for water adsorption.

## 3.5 Particle Formation Modeling

The initial particle formation rate, $J$, defined as the rate at which new particles form under given ambient conditions, is the main quantity characterizing the particle formation process (Yazgi and Olenius, 2023). However, the extent to which hydration influences $J$ remains unknown. Figure 9 shows the enhancement of the particle formation rate, $J/J_{\mathrm{RH=0\%}}$, as a function of relative humidity (RH). For clarity, the figure highlights only the overall range of enhancement, while detailed results for individual simulations are provided in Sec. S8.

Here we only apply the commonly used and well-performing methodology, DLPNO$^{\mathrm{NormalPNO}}$–CCSD(T$_0$)/aug-cc-pVTZ//$\omega$B97X-D/6-31++G(d,p) level of theory with the quasi-harmonic correction applied, which we use to obtain binding free energies. In addition, we also corrected the binding free energies according to the Halonen limit, providing two limiting cases: without any anharmonic corrections (QC+QHA) and with the largest corrections that can be reasonably expected (QC+QHA+HL). Based on the uncorrected QC+QHA data, the MSA–MA and SA–AM systems exhibit modest, positive enhancements, with





$J$ increasing by no more than a factor of two even at high RH. For SA–DMA, hydration can either increase or decrease $J$ depending on the ambient conditions, but the effect remains within a factor of $\sim$2.

When binding free energies are corrected according to the Halonen limit (QC+QHA+HL), we find substantially larger enhancements, with the MSA–MA and SA–DMA systems showing increases of up to a factor of 20. Even so, this corresponds

to only about one order of magnitude under the most favorable conditions. Since evaporation rates depend exponentially on binding free energies, an error of just 3 kcal mol$^{-1}$ can likewise produce a factor of 20 difference in $J$. As the actual anharmonic corrections are expected to represent only a fraction of the Halonen limit, these results suggest that hydration does not substantially influence particle formation, particularly when compared to the current level of computational uncertainty across different computational methods.

We note that, to obtain more accurate particle formation rates, the largest cluster size in the ACDC simulations should ideally be extended, since the critical cluster size may exceed the present cutoff at high temperatures and/or low monomer concentrations.

In comparison with previous studies, our results predict a much smaller enhancement for the MSA–MA system than reported by Chen et al. (2020). At 278.15 K with [MSA] = $10^6$ cm$^{-3}$ and [MA] = 10 ppt, they reported enhancements of approximately

seven orders of magnitude for $J_{4\times4}$ and three orders of magnitude for $J_{2\times2}$, which can be attributed to a large decrease in binding free energy upon hydration of the clusters. To directly assess this, we recalculated the lowest-energy structures reported in their Supporting Information at the same level of theory (DLPNO$^{\text{NormalPNO}}$–CCSD(T$_0$)/aug-cc-pVTZ//M06-2X/6-31++G(d,p)). We obtained binding energies of $-6.25$ kcal mol$^{-1}$ for MSA$_1$MA$_1$ and $-9.34$ kcal mol$^{-1}$ for MSA$_1$MA$_1$W$_1$ at 278.15 K, substantially weaker than the corresponding values of $-7.18$ kcal mol$^{-1}$ and $-13.23$ kcal mol$^{-1}$ reported by Chen et al. (2020).

While minor differences in DLPNO settings could account for small discrepancies, they cannot explain differences of this magnitude. Moreover, our recalculated values are consistent with those obtained using our DLPNO$^{\text{NormalPNO}}$–CCSD(T$_0$)/aug-cc-pVTZ//$\omega$B97X-D/6-31++G(d,p) method (e.g., $-6.53$ kcal mol$^{-1}$ and $-9.40$ kcal mol$^{-1}$, respectively). Taken together, these results suggest that the hydration effects reported by Chen et al. (2020) for the MSA–MA system are substantially overestimated.

Our results for the SA–DMA system are in good agreement with Henschel et al. (2016), who reported enhancements of at most a factor of two for [SA] = $10^5$ cm$^{-3}$ at 263 K. In contrast, Henschel et al. (2016) also found enhancements of up to a factor of 50 for the SA–AM system (263 K, [SA] = $10^5$ cm$^{-3}$, [AM] = 10,000 ppt), which we do not observe under any conditions (even in the Halonen limit used at 263 K).

These comparisons underscore the strong dependence of $J$ on the specific QC calculations employed. To more reliably

assess the effects of hydration, calculations should be performed consistently using the same QC methodology across a range of relevant systems.



## 4 Conclusions

In this work, we systematically benchmarked a broad range of quantum chemistry methods for their ability to describe hydrated atmospheric molecular clusters. The comparison focused on electronic binding energies and equilibrium geometries. Based on

this benchmarking, we identified the most accurate methods and used them to evaluate the magnitude of different thermochemical corrections, thereby improving the binding free energies beyond the conventional approach of applying a harmonic oscillator approximation to a single low-energy structure. Finally, we calculated hydration distributions and particle formation rates using both the uncorrected quantum chemistry binding free energies and the same data adjusted with the maximum correction expected for an ideal system.

The widely used DLPNO$^{\text{NormalPNO}}$–CCSD(T$_0$)/aug-cc-pVTZ//$\omega$B97X-D/6-31++G(d,p) method performed well in the benchmarks and was therefore employed in the thermochemical analysis as well as in the hydration distribution and particle formation rate calculations, for consistency with previous studies. We also identified $\omega$B97X-3c as an efficient and accurate option for large-scale studies, with or without electronic energy corrections, while DLPNO$^{\text{NormalPNO}}$–CCSD(T$_0$)/aug-cc-pVTZ and LNO$^{\text{Tight}}$–CCSD(T)/aug′-cc-pVTZ provide accurate single-point electronic energy corrections. The low computational cost

and memory requirements of the LNO$^{\text{Tight}}$–CCSD(T) methods are particularly noteworthy, and we recommend them for future studies. For very large clusters, B97-3c remains computationally practical, though we advise correcting its electronic energies with LNO$^{\text{Tight}}$–CCSD(T) (Knattrup and Elm, 2025).

We provided a general overview of methods for improving the thermochemical description of molecular clusters, including treatments of vibrational anharmonicity, low-frequency modes, and multi-conformational contributions, each of which can

introduce significant corrections to the free energies of hydrated clusters. However, assessing the accuracy of these corrections remains challenging due to the limited availability of experimental reference data. To complement the quantum-chemical approaches, we employed umbrella sampling, which yielded substantially lower binding free energies, approaching the theoretical upper bound on stabilization for ideal systems reported by Halonen (2024). Given the difficulty of determining which methodology provides the most reliable description, we considered a systematic range of cluster free energies and incorporated

this uncertainty into the hydration distribution and particle formation rate calculations.

Our results show that the hydration distributions of atmospheric acid–base clusters depend strongly on both cluster composition and the choice of computational method. Because these hydration trends are method-dependent, maintaining consistency in the chosen method is crucial when comparing hydration distributions. Under atmospheric conditions, most monomers and dimers remain largely unhydrated, although dimers containing sulfuric acid or methanesulfonic acid exhibit a greater tendency

to bind water. Larger clusters display a pronounced increase in hydration capacity, though the extent depends on factors such as molecular symmetry, available hydrogen-bonding sites, and the presence of hydrophobic alkyl groups. Applying Halonen's upper thermodynamic limit further increases the predicted water content of molecular clusters, though not uniformly across all systems.

When translated into particle formation rates, hydration was found to exert only a modest influence. For sulfuric acid–

ammonia, sulfuric acid–dimethylamine, and methanesulfonic acid–methylamine systems, the enhancement of new particle

formation (NPF) due to humidity rarely exceeded a factor of $\sim$1–2 under typical atmospheric conditions. In some cases, hydration even suppressed growth by preferentially stabilizing smaller clusters over larger ones. Applying systematic thermochemical corrections (up to the Halonen limit) amplified the effect of hydration, but only up to about one order of magnitude. These results indicate that hydration has a relatively minor thermochemical impact on the earliest steps of NPF. However, this does not imply that humidity is unimportant for later growth, where higher relative humidity may reduce the sticking probability of incoming vapors and where the formation of surface layers at the aerosol–air interface could influence uptake and stabilization of additional molecules. Future work should extend these benchmarks to larger clusters and later growth stages, where humidity is expected to play a more pronounced role in aerosol evolution.

Comparing to previous studies, our results largely agree with Henschel et al. (2014) regarding the role of hydration in sulfuric acid–ammonia and sulfuric acid–dimethylamine systems. However, the strong humidity dependence reported for methanesulfonic acid–methylamine by Chen et al. (2017) appears to be overstated. While systematic errors in statistical thermodynamics, as discussed by Halonen (2024), do influence absolute predictions of new particle formation rates, their impact on relative humidity-driven enhancements appears to be minor. Nonetheless, the uncertainties associated with statistical thermodynamics following quantum-chemical calculations warrant further attention in future studies.

*Data availability.* All the calculated structures and thermochemistry are available in the Atmospheric Cluster Database (ACDB) at: https://github.com/elmjonas/ACDB/tree/master/Articles/neefjes25_hydration

*Author contributions.* Conceptualization: JE, JK; Methodology: IN, HW, JE, JK; Software: JK; Validation: IN, YK, HW, JK; Formal analysis: IN, YK, HW, JK; Investigation: IN, YK, HW, JK; Resources: JE, JK; Data Curation: IN, YK, HW, GBT, JK; Writing - Original Draft: IN, JK; Writing - Review & Editing: IN, YK, HW, GBT, JE, JK; Visualization: IN, JK; Supervision: JE, JK; Project administration: IN, JK; Funding acquisition: JE, JK;

*Competing interests.* At least one of the (co-)authors is a member of the editorial board of Aerosol Research. The authors have no other competing interests to declare.

*Acknowledgements.* Funded by the European Union (ERC, ExploreFNP, project 101040353 and MSCA, HYDRO-CLUSTER, project 101105506) and the Danish National Research Foundation (DNRF172) through the Center of Excellence for Chemistry of Clouds. Views and opinions expressed are however those of the authors only and do not necessarily reflect those of the European Union or the European Research Council Executive Agency. Neither the European Union nor the granting authority can be held responsible for them. The numerical results presented in this work were obtained at the Centre for Scientific Computing, Aarhus https://phys.au.dk/forskning/faciliteter/cscaa/.





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
