# Peer review of "Thermodynamic Benchmarking of Hydrated Atmospheric Clusters in Early Particle Formation"

_Aerosol Research, 2025_

## Author Comment (AC1)

**Response to Peer Reviews**

We sincerely appreciate the constructive and encouraging remarks from all three referees. All the issues raised have been carefully considered, and the corresponding corrections have been incorporated into the revised manuscript. We hope that the explanations provided below adequately address each point, and that the manuscript is now deemed suitable for publication. The referees' comments are presented in blue, followed by our detailed response to each one. Excerpts illustrating revisions to the article use green to mark newly added text and  to indicate removed text. Page and line numbers refer to the track-changed revised manuscript. Nevertheless, we would first like to comment on (minor) corrections in our umbrella sampling simulations.

**########## ERRATA**

After submission, we discovered that a bug in our post-processing code had introduced an error in the umbrella-sampling post-analysis, affecting the values shown in Figure 5. The temperature values were accidentally exchanged with molecular mass values for the $SA_1DMA_1W_{0-4}+W_1$ bindings. We have corrected this issue and updated the figure accordingly. This correction does not affect any other results or the overall conclusions of the manuscript. In addition, we removed the QHA contribution from the QHA+US correction, treating the US correction as applied on top of QC rather than QC+QHA, which is more reasonable. The revised umbrella-sampling estimates now fall within a physically reasonable range, while still approaching the Halonen limit. This section thus now more reasonably demonstrates that entropic contributions to the Gibbs free energy may be substantially larger than those captured by standard computational thermodynamic treatments, underscoring the potential magnitude of thermochemical errors in atmospheric cluster studies. We have also made small adjustments to the text to reflect these corrections.

[Figure]

**Change on Line 409:** "As shown in Fig 5, the  US

corrections on top of the QC calculations (red)  yields values close to the Halonen limit, suggesting that the traditional QC approach may significantly underestimate cluster binding free energies, and the  thermodynamic corrections indeed might be close to the Halonen limit."

########## RC1 ###########

**RC1:** The authors conducted a systematic study of cluster hydration in atmospherically important systems. This purely theoretical study primarily focuses on quantum chemical approaches and evaluates them against one another. Such benchmarking is standard yet crucial for future studies. Additionally, the authors enhance their investigations by exploring cluster properties beyond the standard quantum chemical framework. I commend the authors for their efforts in this regard!

The benchmarking is extensive and done with high rigor. The group's ability of carrying out such method comparisons is well known and overall top quality, and I have very little to comment or criticize about.

A general remark: While the authors note that no good reference data exist for cluster thermodynamics, I find it useful and interesting that they investigate the potential effects of anharmonicities. These systems likely exhibit significant thermal fluctuations, making the standard harmonic approach inadequate. Therefore, I believe the results and speculation are sufficient as they are and are generally highly welcome.

**Authors:** We thank the referee for the positive and supportive comments.

**RC1:** The nucleation rate section (3.5) is clearly the weakest part of the study, which is unfortunate because it discusses the real-life implications. I will elaborate on my critique below:

In the ACDC simulations, the limiting size is set to 3 acids, 3 bases, and any number of water molecules. Given the studied concentrations, the system size is too small. Except for a few extreme cases, the critical cluster is not included in the set of studied clusters. This is briefly mentioned on Line 465, but not very clearly. Additionally, in the supplementary information, they state:
"Because the maximum simulated cluster size was relatively small, the critical cluster size, where growth starts to vastly outweigh evaporation, might not be included" and "Elm et al.(6) coined the term: potential particle formation rate Jpotential, to indicate that the results cannot directly be related to the actual particle formation rate J, but are rather a measure for the importance of different compounds in cluster formation."

**Authors:** We thank the referee for this thoughtful critique. We fully agree that the limited cluster-size range means that the critical cluster is not included for most conditions, and thus the ACDC outputs represent potential particle formation rates. We have now clarified this explicitly and prominently in the main text to avoid any misinterpretation.

**Change on Line 487:**  We note that obtaining quantitatively accurate particle formation rates would require extending the maximum cluster size in the ACDC simulations, as the critical cluster is likely larger than our current cutoff under many of the studied conditions. Consequently, the rates reported here should be interpreted as potential particle formation rates rather than true nucleation rates.(Elm et al., 2021) Nevertheless, we emphasize that our aim is not to determine absolute/potential nucleation rates, but to assess the relative effect of hydration. Because we compare hydrated and dry simulations under otherwise identical conditions, systematic uncertainties largely cancel, making the resulting rate ratios less sensitive to the limited cluster-size cutoff. Thus, while the absolute rates are not quantitative, the relative impact of humidity is more robust.

Despite this limitation, the simulations still allow us to robustly assess the relative effect of hydration on the earliest steps of cluster formation. Across all studied systems, hydration modifies the rate only within roughly one order of magnitude; substantially smaller than typical uncertainties arising from quantum-chemical energetics or thermochemical corrections. Thus, while the absolute rates cannot be interpreted as true nucleation rates, the qualitative conclusion remains well supported: hydration has a limited influence on the very first steps of cluster formation under the studied atmospheric conditions.

**RC1:** And the article cited in the SI (Elm et al., 2017) does not include the concept or its coining. It is quite frustrating that authors miscite, but come on, it's one of your own!

**Authors:** We thank the reviewer for catching this miscitation. The concept indeed comes from Elm et al. (2021), not Elm et al. (2017). We have now corrected the citation in the SI.

Citation corrected.

**RC1:** If nothing else, I strongly think the authors should at least explicitly state that the nucleation rates studied are not true nucleation rates. Currently, the results may mislead readers into thinking otherwise. As noted in the SI, the measure used here is the potential particle formation rate, which could serve as a good proxy. However, given the magnitude of the effect of hydration (within one order of magnitude), this analysis needs to be more rigorous.

**Authors:** We thank the reviewer for this comment. We fully agree that it is important to state clearly that, in most cases, our reported values represent *potential* particle formation rates rather than true nucleation rates. We now emphasize this explicitly in the main text to avoid any misunderstanding (see our response to the second comment). We also agree that a more rigorous treatment would be required to obtain quantitatively accurate humidity-dependent NPF-enhancement factors. However, many other aspects of the

modeling framework and the underlying quantum-chemistry calculations would likewise need substantial refinement before such values could be interpreted in an absolute sense. Within this context, our conclusion, that humidity affects particle formation by less than one order of magnitude, remains meaningful, particularly given that some previous studies suggested a strong stabilizing effect of water on cluster thermodynamics, an effect not supported by our results. The corresponding corrections to the manuscript are detailed in our response to the second comment.

**RC1:** Let me elaborate on my point: The idea that incomplete cluster formation free energies can indicate the magnitude of the nucleation rate applies when comparing nucleation capabilities among different chemistries (e.g., SA-AM to SA-DMA). However, in this case, the comparison is made against hydration or, more precisely, relative humidity. The authors demonstrate, particularly in fig. 7 and 8, that the level of hydration depends on cluster size in a non-monotonic manner. Therefore, extrapolating the level of hydration to larger, unknown clusters lacks proper justification.

**Authors:** This is precisely why we avoid extrapolating hydration patterns to larger, unsampled clusters and refrain from interpreting our results as absolute nucleation rates. Although hydration varies non-monotonically with size, several physical arguments (e.g., Fig. 8) indicate that the overall magnitude of the humidity effect is unlikely to increase abruptly by orders of magnitude at larger sizes. While we agree that the non-monotonic behavior prevents direct extrapolation, we expect the general trend to persist.

**RC1:** To provide a more theoretical perspective: The nucleation rate largely depends exponentially on the free energy of the critical cluster only: $J \sim exp(-dG^*/RT)$. Thus, $J$ primarily depends on the height of the nucleation barrier, not its slope. Consequently, if hydration significantly decreases free energy at the critical size but has smaller effect around it, the nucleation rate will increase notably and "unpredictably". The same reasoning applies in reverse if the critical cluster is hydrophobic among otherwise hydrophilic clusters.

**Authors:** We do not compare the slope, but indeed, we try to assess the change of the barrier height from the changes observed in free energies of critical clusters or close-to-be-critical clusters (i.e., the largest clusters in the simulation). This is fair, as within the explicitly sampled size range, hydration effects remain modest and relatively smooth, and our analysis focuses solely on the relative impact of hydration between hydrated and dry simulations under identical conditions. While the behavior of larger clusters cannot be ruled out as more complex, the ratio-based comparison we perform still provides a robust indication that hydration does not dramatically alter the early stages of cluster formation in the studied systems.

**RC1:** I really don't understand why the authors chose to study these concentrations. Since they are not directly comparing their results to any experimental or measurement data, they could select conditions where the critical cluster appears within the set of clusters. I believe

that limiting the range of critical clusters is a lesser issue than simulating incomplete systems. I kindly suggest that they consider running the simulations at other concentrations.

**Authors:** We thank the reviewer for this suggestion. The concentrations used in our ACDC simulations were chosen because they represent typical atmospheric levels of the acids and bases considered here. Our goal in this section is to evaluate the relative influence of hydration under realistic atmospheric conditions. For this purpose, it is important that the concentrations correspond to those observed experimentally in the atmosphere. However, we agree that, at these concentrations, the critical cluster may not always lie within the simulated size range. However, as discussed in our responses above, the quantity we focus on—the potential effect of humidity on particle formation—primarily reflects relative changes in stability, which are much less sensitive to where exactly the critical cluster falls. Tests with alternative conditions support that the qualitative impact of hydration remains within the same order of magnitude.

For these reasons, we prefer to retain the atmospheric concentrations used here, as they best represent the conditions under which hydration is expected to be relevant.

**RC1:** Some minor comments and questions:
Line 207: Coagulation loss is referred to as CL, but it is also CS (SI, p. S6). Should CL/CS have a unit?

**Authors:** We thank the reviewer for noting this inconsistency. We have corrected the notation so that coagulation loss is referred to consistently as CL (i.e., not CS initially used for coagulation sink). CL indeed has units of $s^{-1}$, which we also corrected.

**Change on Line 212:** "Coagulation loss (CL) of clusters was included using CL = $10^{-3}$ $(d/d_{SA})^{-1.6}$ $s^{-1}$, …"

**Change on Page S6:** "We have used C̶S̶L$_{ref}$ = $10^{-3}$ $s^{-1}$, …"

**RC1:** Line 306: The authors state that they have considered multi-conformer Boltzmann averaging, as explained later in Section 3.3.3. However, eq 4 does not resemble Boltzmann averaging. Could the authors elaborate on this approach?

**Authors:** We thank the reviewer for this comment. We apply proper Boltzmann averaging over energy states of multiple low‑lying conformers; however, Eq. 4 is not the Boltzmann average itself but the resulting free-energy correction after the Boltzmann weighting. We agree that we could be more careful in clarifying this.

**Added on Line 393:** Although Eq. (4) is written compactly, it carries out full Boltzmann averaging over all conformer energy states: each conformer is weighted by $e^{-\Delta\Delta G_i/k_B T}$, and the logarithmic form simply converts this conformer-weighted partition sum into the corresponding multi-conformer free energy.

**RC1:** Figure 8 is somewhat difficult to read because of the various colors, markers, line types, and shaded areas. I suggest that the authors consider dividing the data into three separate plots or finding an alternative way to present it for maximum clarity. Otherwise, the visual elements of the manuscript are clear and well presented.

**Authors:** We are happy that most visual elements of the manuscript are clear and thank the reviewer for the suggestion regarding Fig. 8. While Fig. 8 is visually dense, the key trends remain clearly visible, and separating it into multiple panels would make cross-comparison more difficult and increase the number of figures considerably. For clarity and coherence, we therefore keep the data presented together in one figure rather than splitting it.

**RC1:** L460: "Since evaporation rates depend exponentially on binding free energies, an error of just 3 kcal mol$^{-1}$ can likewise produce a factor of 20 difference in J." Is this correct? I would assume it is 3RT, approximately 1.8 kcal/mol: exp(1.8 kcal/mol / RT) = exp(3) = 20.

**Authors:** We thank the reviewer for pointing this out. The reviewer is correct that an energy difference of approximately 3 $k_B T$ (≈1.8 kcal/mol at 298 K), not 3 kcal/mol, leads to a factor of ~20 in the evaporation rate via exp($\Delta G_{error}/k_B T$). We have corrected this statement in the manuscript accordingly.

**Change on Line 483:** "... 3  $k_B T$ (≈1.8 kcal mol$^{-1}$ at 298 K) can likewise produce a factor of 20 difference in $J$."

########## **RC2** ###########

**RC2:** The manuscript by Neefjes et al. evaluates the accuracy of quantum chemical methods in predicting electronic binding energies and equilibrium geometries of clusters containing various atmospheric acids, bases, and up to five water molecules. They identify optimal methods for different purposes, use the most reliable methods to investigate the magnitude of thermochemical corrections (anharmonicity, multi-conformer contributions) and, ultimately, to calculate clusters hydration distributions and particle formation rates (J) for three key systems: SA–AM, SA–DMA, and MSA–MA. A key finding is that while hydration does occur and can stabilize clusters, its effect on enhancing NPF rates is generally modest (typically less than a factor of 2) under typical atmospheric conditions, contradicting some previous studies that reported very large enhancements. The work is methodologically sound, and the conclusions are well-supported by the data. It provides invaluable practical guidance for the aerosol community and further clarifies long-standing questions regarding the role of humidity in new particle formation (NPF). I recommend publication in *Aerosol Research* after the following few minor comments have been addressed.

**Authors:** We thank the referee for their accurate summary of our work and their positive overall assessment. We appreciate the constructive feedback and address each of the minor comments below.

**RC2:** 1. The authors correctly note in the conclusion that humidity may play a more significant role in later growth stages (e.g., for larger clusters or aerosol particles), and this distinction could be emphasized earlier to provide context.

**Authors:** We agree with the referee that this is an important distinction. To make the scope of the study clear from the outset, we have added a clarification in the Introduction, noting that our analysis focuses on the initial stages of new particle formation.

**Change on Line 82:** A growing body of computational studies on atmospheric clusters increasingly supports the systematic inclusion of water in cluster modeling. To facilitate this integration, we benchmark quantum chemistry methods for their accuracy in describing hydrated clusters. This study specifically focuses on the initial stages of new particle formation, involving freshly nucleated particles a few nanometers in size, whereas hydration effects in subsequent growth stages may differ substantially. We evaluate key properties such as binding electronic energies, cluster geometries, vibrational frequencies, and binding free energies. Furthermore, we analyze the hydration distributions across different cluster sizes and compositions and examine the cluster distribution dynamics of the most relevant systems. Thus, this work not only assesses the accuracy of current methods in describing hydrated clusters but also reveals how explicitly incorporating water can influence conclusions regarding the role of humidity in NPF.

**RC2:** 2. In the methodology, considering the "up to five distinct low-energy configurations", what would be the potential for missing significantly populated conformers in this sampling strategy?

**Authors:** We thank the reviewer for this question. We would like to clarify that the 'up to five low-energy conformers' apply only to the benchmarking sections, where our goal is to work with a representative set of distinct, well-defined minima. For benchmarking electronic structure methods, geometry optimizations, and vibrational frequencies, a small but diverse set of conformers is sufficient and avoids unnecessary duplication of very similar structures. Therefore, we expect negligible errors in our sampling approach for this purpose.

For the thermochemistry analysis, however, we use a different and much broader configurational sampling procedure. Here, the aim is not representativeness but completeness of the low-energy landscape, and therefore many more structures are generated, optimized, and evaluated. The thermochemical results do not rely on the five-structure selection used in the benchmarking part.

**RC2:** 3. Page 12, Line 280: "ωB97X-3 could serve as an efficient method..." – There is a typo here, it should be "ωB97X-3c".

**Authors:** We thank the referee for spotting this typo and have corrected it accordingly.

**Change on Line 285:** "ωB97X-3c"

**RC2:** 4. Page 2, Line 59. The sentence "In recent decades, computational…" deserves few more citations

**Authors:** The referee's point is well taken. The substantial contributions of computational studies to understanding new particle formation merit a broader set of citations. Accordingly, we have added additional references and included an "e.g." to indicate that this list is non-exhaustive.

**Change on Line 59:** "In recent decades, computational chemistry methods have been extensively used to address this challenge (e.g., Vehkamäki et al., 2002; Nadykto and Yu, 2007; Kurtén et al., 2008; Temelso et al., 2012; Almeida et al., 2013; Liu et al., 2018; Xu et al., 2020; Elm et al., 2020)"

**RC2:** 5. What do the authors mean by "Intrinsic basis set" and "User-supplied basis set" in Table 1, while empirical, semi-empirical and other methods are listed instead? Consider providing clarifications or renaming the headers.

**Authors:** The intention behind this classification was to distinguish methods with a built-in, fixed basis set from those for which the basis can be chosen by the user. For instance, GFN1-xTB employs a minimal Slater-type basis that is inherent to the model and cannot be modified, whereas functionals such as ωB97X-D can be paired with different user-selected basis sets. This distinction was meant to clarify that the basis functions listed in the third column apply only to the methods under "user-supplied basis set." The referee is correct that these terms were not standard, so the column headers have been revised for clarity, and additional explanation has been added to the caption.

**New headers:**    Fixed internal basis set – User-selected basis set

**Caption:** Overview of the quantum-chemistry methods and basis sets included in this benchmark . Methods are grouped according to whether they use a fixed internal basis set or require a user-selected basis set. For methods requiring a user-selected basis set, one or more of the basis sets listed in the final column were used.

**RC2:** 6. The discussion of the "Halonen limit" is excellent for setting an upper bound, but the manuscript could more clearly state what the authors believe is the most likely realistic scenario based on their umbrella sampling and anharmonicity analysis. Is the truth closer to the QHA result or the Halonen limit?

This is an interesting and important question. Our updated umbrella-sampling results (see the errata above) yield binding free-energy corrections that are slightly lower in magnitude, but close to the Halonen limit. This suggests that, for the systems studied here, the QHA approach indeed neglects significant entropic contributions.

At the same time, applying umbrella sampling to obtain binding free energies for atmospheric clusters is still a very recent development. This application was, to the best of our knowledge, first introduced in our publication Kubečka et al. (2025;

10.1021/acsomega.5c05634). As a result, there is not yet a sufficiently broad dataset to establish whether corrections close to the Halonen limit are characteristic only of certain cluster types or represent a more general trend. In our previous study, for instance, we observed a wide spread of free-energy correction magnitudes even for dimer formation.

We also emphasize that, because of the high computational cost of first-principles umbrella sampling, we rely on machine-learning potentials trained on gradient calculations evaluated at GFN1-xTB geometries. Although we apply rigorous error analysis, this necessarily means that the sampling explores the GFN1-xTB potential-energy surface, which may introduce some uncertainties into the resulting high-level free-energy profiles.

These points do not reflect a lack of confidence in our methodology. On the contrary, our current results consistently indicate that QHA can substantially underestimate the true Gibbs free energy for the clusters studied here. Rather, we aim to avoid overgeneralizing at this early stage of applying umbrella sampling to atmospheric clusters. Our primary goal in the present manuscript is to highlight that potentially large deviations from QHA may have significant implications for predicted cluster stabilities and particle-formation rates, and that a more complete dataset will be essential for establishing how widespread this effect is.

########## RC3 ###########

**RC3:** The manuscript presents an extensive benchmark study of quantum chemical methods for the calculation of structures and energies of molecular clusters of atmospheric relevance. Furthermore, it evaluates the impact of various thermochemical corrections on cluster energies and distribution of hydrates. Finally, the results are applied in the calculation of particle formation rates. The overall quality of the manuscript is good, and the data is clearly of relevance. However, as the focus of the study is on benchmarking different methods for energy calculations, whereas calculations of particle formation rates only represent a minor part of the manuscript with less novel insight, the authors might want to adjust the title of the manuscript to reflect this.

**Authors:** We agree that the main focus of the work lies in benchmarking quantum chemical methods, with particle formation rate calculations playing a supporting role. Accordingly, we have revised the title to better reflect the scope of the manuscript.

**From:** Effect of humidity on the first steps of atmospheric new particles formation: Computational study of hydrated molecular clusters

**To:** Thermodynamic Benchmarking of Hydrated Atmospheric Clusters in Early Particle Formation

**RC3:** In the description of the calculations, it is somewhat unclear at which levels of theory geometry optimization were performed. At the end of page S2 it is stated that the generated

cluster geometries are "used as a starting point for geometry optimizations with the quantum chemistry methods included in the benchmark." This would include all methods used in the electronic binding energy benchmark section. If that is the case, why were not all these model chemistries included in the equilibrium geometry benchmark? In case geometries were not optimized at all these levels, this should be clarified. In that case I would also suggest reorganizing the manuscript by swapping places between the two sections, as the method recommended based on the equilibrium geometry benchmark is used to generate the structures used for the electronic binding energy benchmark.

**Authors:** We thank the reviewer for this detailed comment. To clarify: although the wording at the end of page S2 may have suggested otherwise, we did not perform geometry optimizations with all electronic-structure methods included in the binding-energy benchmark. We acknowledge the reviewers' comment that the choice of geometries/optimization QC methods is not clear in all sections of our manuscript. Therefore, we have revised the text to make this workflow explicit and avoid any implication that geometry optimizations were performed at all benchmarked levels.

2.3 Electronic binding energy benchmark
**Change in Line 154:** "Using both the microhydrated monomer and dimer clusters and (sulfuric acid–ammonia)-pair clusters, with geometries optimized at the GFN1-xTB level of theory (see Section 2.1), all QC methods were benchmarked based on their electronic binding energy $\Delta E_{el}$ for the given geometries"

**Change in Line 178:** "All sampled  (GFN1-xTB-equilibrium) geometries  successfully reoptimized at ωB97X-D/6-31++G(d,p) level of theory (see Sec. S1) were further reoptimized with each benchmarked method."

**Change on Page S2:** "... , resulting in a final dataset consisting of 1,831 non-equilibrium structures (though equilibrium at the GFN1-xTB level), and these structures were used for the electronic energy benchmark."

**Change on Page S3:** "In the end, 1,831 structures remained, which were then used as the starting point for geometry optimizations with the quantum chemistry methods included in the equilibrium geometry benchmark."

**Change on Page S3, end of "Microhydrated (sulfuric acid–ammonia)-pair clusters" subsection:** These structures were further used in the electronic energy benchmark.

**RC3:** The authors seem to misunderstand or at least misrepresent the way that scaling factors for calculated vibrational frequencies are commonly used. Scaling factors are described to be used to adjust for deviations of harmonic frequencies from their anharmonic counterpart, whereas they are indeed generally used to adjust calculated frequencies (harmonic or anharmonic) to match experimental frequencies. Although the lack of

anharmonicity is often a main contribution to the deviation, it is not necessarily dominant.

**Authors:** We thank the reviewer for this clarification. We agree that vibrational scaling factors are often introduced in the literature as empirical corrections intended to bring calculated frequencies into agreement with experimental values, and therefore account for several sources of error (anharmonicity, basis-set effects, and method deficiencies). Our wording focused on the computational perspective, where scaling factors are commonly applied as a practical approximation for anharmonic corrections. This approach is particularly useful when explicit anharmonic frequency calculations are too computationally expensive. To avoid confusion, we have revised the text to acknowledge both viewpoints. In the revised manuscript, we also clarify that our use of scaling factors follows an established strategy in cluster vibrational spectroscopy, where harmonic frequencies are fitted to VPT2 anharmonic values when experimental fundamentals are not available, as demonstrated for water clusters by Temelso et al. (J. Phys. Chem. A 2011, 115, 12034–12046).

**Change in Line 338:** "However, due to practical limitations such as computational cost and convergence issues, vibrational scaling factors are often applied as a simpler alternative. While scaling factors are commonly derived empirically to improve agreement with experiment and thus correct several sources of systematic error (e.g., missing anharmonicity, basis-set incompleteness, and method deficiencies), they are also frequently used as a practical approximation to anharmonic corrections when explicit anharmonic calculations are not feasible. In this work, we follow the latter strategy by deriving scaling factors from comparisons between harmonic and VPT2 anharmonic frequencies, consistent with established practice in cluster studies (e.g., Temelso et al., 2011)."

**Change on Page S16:** "As a more practical alternative to full anharmonic treatments, vibrational scaling factors are applied to approximate anharmonic corrections when explicit VPT2 calculations are computationally demanding. Here, due to the lack of experimental vibrational data for these clusters, the scaling factors are derived by comparing harmonic and VPT2 anharmonic frequencies."

**RC3:** In this context the authors also somewhat misrepresent the study by Jacobsen et al. (2013) (line 334-336), suggesting the study's conclusion to be that anharmonic calculations can be replaced by scaling factors, whereas the study actually compares scaled harmonic frequencies to scaled(!) anharmonic frequencies.

**Authors:** We thank the reviewer for the comment. We agree that Jacobsen et al. (2013) compared scaled harmonic with scaled anharmonic frequencies. Their specific conclusion was that, for the small basis sets and methods they examined, scaled anharmonic frequencies showed no significant improvement over scaled harmonic ones. While we are aware of this, we agree that it is not clear from the text and thus we have revised the text to reflect this more accurately.

**Change in Line 344:** "Jacobsen et al. (2013) further note that  for the small basis sets and methods they tested, scaled anharmonic vibrational frequencies were not significantly more accurate than scaled harmonic ones when compared with experiment, indicating that explicit anharmonic calculations provided limited additional accuracy in that regime."

**Change on Page S16:** ""

**RC3:** Given this context, the authors' approach to determine scaling factors using anharmonic frequencies calculated at the same level of theory as reference makes limited sense. Certainly, as we know from Jacobsen et al. (2013) that only calculating anharmonic frequencies using simplified model chemistries might not be sufficient to reproduce experimental spectra. A better approach would be to make use of the experimental data that the authors have gathered and determine scaling factors based on those.

**Authors:** We thank the reviewer for this thoughtful comment. We agree that, in principle, the most rigorous way to derive vibrational scaling factors is to fit calculated frequencies to experimental fundamentals. However, for the hydrated clusters studied here, reliable experimental vibrational assignments are limited: many bands are strongly broadened, overlapping, or ambiguous, and several species do not have experimentally resolved fundamentals at all. This prevents the construction of a consistent, cluster-specific experimental benchmark suitable for deriving scaling factors. For this reason, we follow a strategy commonly used in cluster vibrational studies: deriving scaling factors by comparing harmonic frequencies with higher-level VPT2 anharmonic frequencies computed at the same level of theory. This approach ensures internal consistency across all cluster types and avoids bias from incomplete or uncertain experimental datasets. Importantly, we do not claim that these scaling factors reproduce experimental spectra, but rather that they provide a practical and internally consistent approximation to anharmonic corrections under conditions where experimental reference data are insufficient. We, however, agree that it should be mentioned in the main text.

**Change in Line 359:** "… applying a single scaling factor. Ideally, scaling factors would be fitted directly to experimental fundamentals, but the lack of consistent, unambiguous vibrational assignments for many of the studied clusters makes this infeasible; we therefore adopt the internally consistent and fully automatable harmonic–VPT2 approach."

**RC3:** It should also be mentioned that scaling factors for r2SCAN-3c have previously been determined (Tikhonov et al. (2024), https://doi.org/10.1002/cphc.202400547).

Thank you for bringing this reference to our attention. We were not aware of this study. A comparison between the absolute scaling factor of 0.9688 reported by Tikhonov et al. (2024) and the single scaling factor of 0.950 obtained in our work for $r^2$SCAN-3c has now been added to Section 3.3.2:

**Change on Line 362:** The scaling factor of 0.950 for $r^2$SCAN-3c is slightly lower than the absolute scaling factor of 0.9688 reported by Tikhonov et al. (2024), a difference that lies within the expected variation when scale factors are fitted to different benchmarking sets. For reference, the two factors differ by more than our MAE of 52 cm$^{-1}$ only for harmonic frequencies above roughly 2766 cm$^{-1}$. It is also worth noting that our two-region scaling (see Sec. S6.4) yields a factor of 0.969 for modes below 2000 cm$^{-1}$, essentially identical to the 0.9688 reported by Tikhonov et al. (2024).

**Change on Page S17:** "Tikhonov et al. (2024) reported a scaling factor of 0.9688. However, at the time of this study, no scaling factors were available for other DFT-3c methods. Therefore, and to assess whether scaling factors differ for cluster systems,  we calculate our own scaling factors."

**Change on Line 349:** Scaling factors have been defined for various methods (Johnson, 1999; Myllys et al., 2016, Tikhonov et al., 2024)

**RC3:** Section 3.3.3: It would be interesting to know how the number of conformers that are taken into account changes with cluster size, as this could be a relevant factor for the size of the correction.

**Authors:** We thank the reviewer for this insightful comment and fully agree that the number of contributing conformers influences the magnitude of the Boltzmann correction. Our initial decision not to include such a table stemmed from the fact that the configurational sampling available for this large set of clusters is not exhaustive enough to yield a reliable or systematically comparable count of distinct minima across all cluster sizes. Because defining and enumerating unique minima in flexible clusters is intrinsically challenging, we were concerned that reporting these numbers might convey a misleading sense of precision. Nevertheless, in order to address the reviewer's request and improve transparency, we now provide the corresponding table in Sec. S7.

**Change in Section 3.3.3:** "We, therefore, calculated the multi-conformational free energy contributions using Eq. 4 at ωB97X-D/6-31++G(d,p) for all clusters fulfilling $SA_{0-3}AM_{0-3}W_{0-8}$. See Sec. S7 for more details on the number of minima used. As shown in Fig. 5, the resulting corrections are relatively small, never exceeding −2 kcal mol$^{-1}$."

In the SI, we added Sec. S7 and updated the numbering accordingly. This new section includes a table listing all clusters, the number of minima used, the number of atoms in each cluster, and the applied/resultant free-energy corrections, e.g.:

| cluster | # | $N_A$ | $\Delta\Delta G$ |
|---|---|---|---|
| 1am | 1 | 3 | 0. |

**RC3:** Section 3.3.4: It seems that the authors do not fully trust their own methodology - the corrections determined by their umbrella sampling approach clearly exceed the limit determined by Halonen in several cases (Figure 5). Yet the authors decide not to use the umbrella sampling results, but instead use the limit as an upper bound. As it is not used or discussed any further, it is not clear why this section is needed.

**Authors:** We thank the reviewer for this comment. After submission of the manuscript, we re-examined our umbrella-sampling calculations and discovered an error in our initial post-processing, which led to an overestimation of the thermodynamic corrections. After correcting this issue, the umbrella-sampling results fall much closer to the Halonen limit (Figure 5), and no longer exceed it. We have updated the analysis accordingly (see beginning of this response letter).

Although the corrected umbrella-sampling values now behave more reasonably, we agree that some uncertainty remains due to the exploratory/pioneering nature of this approach and the well-known challenges of sampling floppy, highly anharmonic clusters. Moreover, these calculations would be significantly more exhausting to do for a wide range of bindings. For these reasons, we chose not to use the umbrella-sampling values directly in the main thermochemistry sections but instead to interpret them as an indication of the possible magnitude of missing physics in standard harmonic treatments.

We retain this section because it highlights an important conceptual point: even when quantum-chemical frequencies and the thermochemistry are adjusted with various corrections, there might be other entropic effects significantly affecting binding in atmospheric clusters. The umbrella-sampling exercise, therefore, serves as a cautionary benchmark and motivates further development of thermochemistry calculation techniques for such systems.

See the ERRATA at the beginning of this response letter for corrections to the main text related to this reply.

**RC3:** Minor comments:
Figures 1, 6, and S10: The structure of TMA is depicted from an unfortunate perspective making it appear planar and hiding several hydrogen atoms.

**Authors:** It indeed makes sense to adjust the TMA structure so that its three-dimensional geometry and all hydrogen atoms are clearly visible. The structure has been replaced accordingly in Figures 1, 6, and S10.

Figures 1, 6, and S10 adjusted.

**RC3:** Line 529: "Henschel et al. (2014)" should likely be "Henschel et al. (2016)"?

**Authors:** We thank the reviewer for noticing this. The correct citation is Henschel et al. (2016), and we have fixed this in the manuscript.

**Change on Line 555:** Comparing to previous studies, our results largely agree with Henschel et al. (2016)

**RC3:** Section S1: The authors state that "reacted" structures are filtered out. What exactly does this refer to - do proton transfer reactions fall under the definition of "reacted"?

**Authors:** We thank the reviewer for this question. In this context, 'reacted' refers to structures in which an unintended chemical reaction has occurred during the optimization (e.g., fragmentation, covalent bond formation, or other rearrangements that change the molecular composition). Proton-transfer configurations are *not* considered reacted; they are treated as valid conformers and included in the analysis. We have clarified this definition in the SI.

**Changes of all instances on Page S2 and S3:** "…filtering out fragmented/reacted (excluding proton transfers) structures …"

**RC3:** Page S3, bottom: The authors state the structures were "refined" at DLPNO-level of theory. Please specify what is meant by this - single point energy or geometry optimization.

**Authors:** We thank the reviewer for noting this. Here, 'refined' was meant to refer to DLPNO single-point energies on DFT geometries; no DLPNO optimization was done. This is now clarified in the SI.

**Changes Page S3:** "The single point energy of the lowest structure was  corrected  with DLPNO…"